# A practical guide to targeted single-cell RNA sequencing technologies
Giulia Moro ✉, Erich Brunner & Konrad Basler

Current single-cell RNA sequencing (scRNA-seq) methods suffer from biases that restrict the detection of cellular transcripts to 10–40% of total RNAs. This hinders the identification of transcripts of interest. Additionally, information retrieved from most high-throughput scRNA-seq methods is confined to untranslated regions toward transcript ends, resulting in loss of detail in internal regions. In this review, we outline biases in scRNA-seq protocol steps that limit transcript and region detection. We then discuss the advantages and disadvantages of targeted sequencing solutions, grouped into five categories according to the protocol step they target. Finally, we present a decision tree that guides researchers in selecting the most suitable targeted method for their experiment.

Single-cell RNA sequencing (scRNA-seq) has emerged as a powerful tool to record and dissect gene expression of individual cells in tissues. Compared to bulk RNA-seq, where RNAs from all cells in a tissue or population are pooled and processed together, scRNA-seq technologies face a key challenge: retaining information at the individual cell level. This is achieved by tagging all RNAs (or cDNAs) from a single cell with a unique sequence (called an index or barcode), with which the origin of each transcript can be identified after sequencing. To do this, single cells must first be isolated into individual reaction chambers, typically droplets or microwells. In most protocols, the RNAs are then reverse transcribed into cDNA and PCR-amplified to generate sufficient material for high-throughput sequencing (HTS).

Over the past decade, the number of cells that can be sequenced, and the simplicity of scRNA-seq protocols have greatly increased[1–6]. Ideally, every RNA in a single cell would be detected after sequencing. However, available scRNA-seq technologies capture only 10–40% of cellular transcripts[1,5,7–11], resulting in substantial information loss. This major limitation often impacts the detection of transcripts of interest (TOIs), such as cell markers, which are important for assigning cellular identity. Beyond detection of transcripts, the sequences of the RNAs themselves contain additional, highly valuable information for specific research or diagnostic applications. For example, RNA sequence data can reveal the presence of point mutations, splice junctions, fusion breakpoints, or uncover rearrangements at CRISPR target sites. We refer to these features as regions of interest (ROIs).

High-throughput, 3′/5′-based scRNA-seq methods are widely used for their ability to profile large numbers of cells and detect numerous transcripts efficiently. Popular platforms include 10x Chromium[12], DROP-seq[13], inDrop[14], and BD Rhapsody[15], which use barcoded beads (microbeads covered with DNA oligonucleotides containing a set of sequences) to capture and tag mRNA molecules. However, a key limitation of these methods is their focus on transcript ends (either 3′ ends or, with newer developments,

5′ ends[16]), resulting in the lack of data from internal regions where ROIs are typically located.

Full-length protocols address the limitations of ROI detection by acquiring sequence profiles across the entire transcript length. These technologies include short-read based protocols (such as Smart-seq technologies[17–22] and VASA-seq[23]) which fragment transcripts or cDNAs before sequencing, and long-read sequencing technologies from PacBio or Oxford Nanopore Technologies (ONT) that directly sequence long nucleotide stretches[24–27]. However, compared to 3′/5′-based methods, full-length technologies process significantly fewer cells (hundreds versus thousands), revealing an important tradeoff[5]: while they provide comprehensive coverage of internal transcript regions containing ROIs, they do so at the expense of cellular throughput.

The relevance of enhanced TOI and ROI detection in scRNA-seq experiments is emphasized by the numerous solutions developed to address this issue. Collectively known as targeted sequencing approaches, these methods are designed to detect specific ROIs within transcripts or to enrich for TOIs important to determine cellular traits.

Here, we provide a broad overview on the targeted scRNA-seq field. We first examine the biases affecting TOI and ROI detection in standard 3′/5′-based scRNA-seq experiments. We then give a comprehensive overview of targeted methods, highlighting the biases they address, their advantages and limitations. While the focus is on methods using barcoded beads, we also mention plate-based protocols. Depending on the targeting protocol used by the method, we group them into five classes: targeted capture, targeted priming, targeted amplification, dual-targeted polymerase chain reaction (PCR) and probe hybridization. Next, we explore long-read sequencing protocols which leverage targeting for improved transcript detection and characterization. Finally, we highlight targeted technologies in the field of spatial transcriptomics and provide examples of biological applications of targeted technologies. To summarize the review, we provide

Department of Molecular Life Sciences, University of Zurich, Zurich, Switzerland. ✉e-mail: giulia.moro2@uzh.ch

a practical decision tree to guide researchers in selecting the most suitable method for their needs and discuss the future directions of targeted scRNA-seq approaches.

## Biases affecting TOI and ROI detection

The detection of TOIs and ROIs in 3′/5′-based methods is influenced by various biases introduced throughout the scRNA-seq workflow (Fig. 1).

Many of these biases were described for bulk RNA-seq experiments[28] but also apply to scRNA-seq protocols as similar steps are used for the library generation. The specific biases occurring during a typical scRNA-seq workflow are detailed in Fig. 1b and described below, listed based on the protocol step they influence.

We further categorize these biases into three groups in Table 1: those inherent to scRNA-seq protocols, those related to transcript sequence

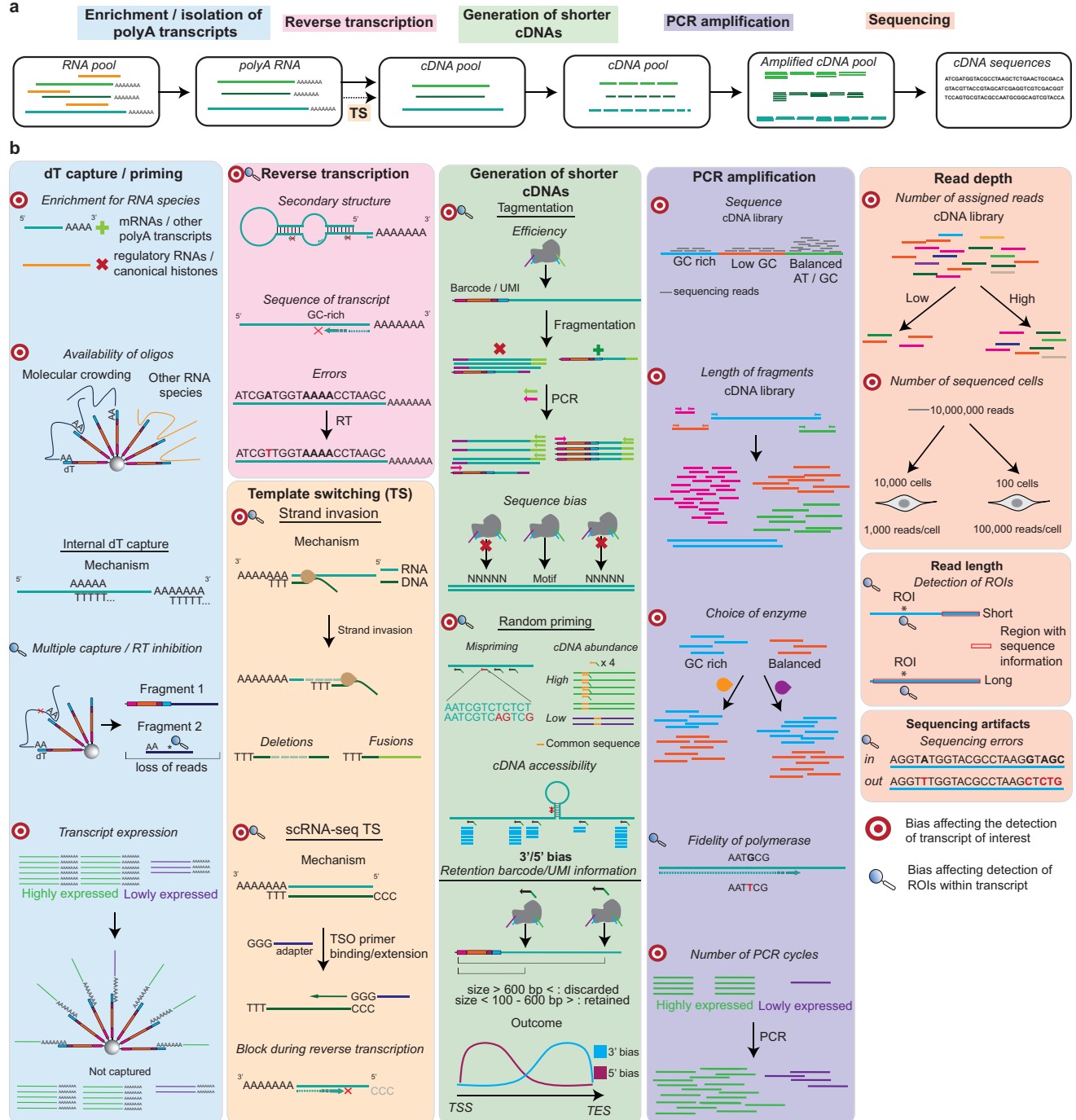

**Fig. 1 | Main steps in 3′/5′-based scRNA-seq experiments and biases affecting each step. a** Most scRNA-seq experiments start with enrichment for polyadenylated transcripts through dT-based capture or priming. RNAs are then converted into cDNAs through reverse transcription, which may include a template switching (TS) reaction. Shorter cDNAs suited for short-read sequencing are then generated, followed by PCR amplification and sequencing. Note: the order of the steps outlined at the top of the figure may vary between scRNA-seq protocols. **b** Bullseyes indicate biases which affect detection of TOIs, while magnifying glasses indicate issues in detecting ROIs. Biases affecting ROI detection are only considered as such if the detection of TOIs is not affected. scRNA-seq TS refers to the reaction in which the terminal transferase activity adds untemplated Cs at the end of the template, followed by recognition by a TSO primer with a complementary stretch of Gs and subsequent extension to generate a second strand.

## Table 1 | Summary of biases influencing the results of 3′/5′-based scRNA-seq experiments

| Bias | Inherent to: | Consequence of bias | Strength of bias for TOI | Strength of bias for ROI |
|---|---|---|---|---|
| dT capture/priming | protocol | loss of information on non-polyA transcripts/loss of information on transcripts | strong | NA |
| Internal dT capture | transcript | loss of reads to non-informative transcripts/incomplete reverse transcription (generation of fragments) | low | low |
| Reverse transcription | transcript | incomplete across full transcript/loss of information on lowly expressed transcripts/sequence errors | strong | strong |
| Template switching | protocol | incomplete across full transcript/fusions and deletions of sequences/sequence artifacts | moderate | moderate |
| Generation of shorter cDNA fragments | protocol | loss of information on coding region/loss of information on cDNAs/long cDNAs (size excluded and lost)/sequence errors/loss of information on lowly expressed transcripts | moderate | high |
| PCR amplification | transcript | loss of information on transcripts/changes in relative abundance of cDNAs/sequencing errors | high | moderate |
| Sequencing depth | experimental design | loss of information on transcripts (especially lowly expressed) | high | high |
| Read length | experimental design | loss of information on regions of interest/assignment of read to transcript | moderate | high |
| Sequencing artifacts | transcript | false conclusions on original sequence of cDNAs | low | moderate |

As the detection of a ROI directly depends on the detection of TOIs, we only consider biases affecting the detection of ROIs if the transcript to which it belongs to is detected. For this reason, the detection of the ROIs not available for the dT capture/priming (NA). The strength of the bias depends on the specific TOI and ROI itself, as well as the protocol conditions used and the experimental design.

characteristics, and those stemming from experimental design choices. Each bias creates limitations affecting TOI and ROI detection to varying degrees, depending on the specific properties of the TOIs and ROIs themselves.

### Preparation of single cells

While often overlooked in scRNA-seq workflows, sample preparation and the isolation of single cells is a critical prerequisite for all experiments[29]. However, these protocols are known to introduce various biases[30]. Although dissociation protocols are tailored to the tissue type, the desired outcome is the same: preserving tissue composition, maintaining intact and viable cells, and minimizing alterations to the transcriptome during processing. Low-quality cells can result in fragmented RNA and an elevated proportion of mitochondrial transcripts, which can dominate sequencing reads and impact TOI and ROI detection. Several strategies have been developed to preserve native gene expression during dissociation, including the use of psychrophilic proteases[31] which help preserve in vivo transcriptional states, or the isolation of single nuclei from tissue (snRNA-seq). Additionally, many scRNA-seq experiments begin with frozen or fixed tissue due to logistical constraints, which can also affect the cellular transcriptome. All of these steps introduce biases which may confound the output of the analysis[30] and impact TOI and ROI detection.

### dT-based capture/priming and internal capture

The aim of most scRNA-seq experiments is the detection of polyadenylated transcripts, primarily protein-coding mRNAs, which constitute only 3–7% of all cellular RNAs by mass[32]. After cell lysis, mRNAs are captured through their polyA-tails, which bind to oligo(dT) stretches on barcoded beads or are targeted using oligo(dT) priming strategies. This polyA-dependent approach is employed by the majority of scRNA-seq methods, with notable exceptions being scFAST-seq[33], MATQ-seq[22], SUPeR-Seq[34] and SMARTer (SMART-seq Total RNA-seq Single Cell)[35] which use alternative strategies.

dT-based capture/priming introduces several biases hampering TOI and ROI detection. One major limitation is the exclusion of non-polyadenylated RNAs, such as regulatory RNAs and canonical histone RNAs. Additionally, although there is a large excess of oligonucleotides on barcoded beads compared to mRNAs expressed in eukaryotic cells ($10^6$–$10^7$ DNA oligonucleotides vs $3$–$10 \times 10^5$ mRNAs[32]), binding may be impaired due to steric hindrance and molecular crowding (where high numbers of macromolecules occupy most of the free space; Fig. 1).

Internal capture artifacts[36,37] further restrict the availability of oligonucleotides on barcoded beads. The same transcript may be captured at different positions, overrepresenting the analyte in the final dataset. Capture of non-polyadenylated transcripts further reduces the pool of oligonucleotides available for binding other transcripts. rRNAs are particularly problematic in this regard, as they constitute 80–90% of eukaryotic samples by mass[32] and can make up 5–75% of reads in scRNA-seq experiments[38,39]. Internal capture also limits ROI detection by generating multiple fragments which are then lost in downstream protocol steps, resulting in a depletion of information at the 5′ end of RNAs[36,37]. A class of TOIs that is particularly impacted is transcription factors, which are notoriously difficult to capture in scRNA-seq experiments[40–42] due to their low expression.

### Reverse transcription

The oligo(dT) sequences on barcoded beads serve two functions: they isolate polyadenylated RNAs and act as primers for reverse transcription. This critical step converts captured RNAs into cDNAs for subsequent PCR amplification but itself introduces several biases[43,44]. Importantly, a one-size-fits-all type of reaction is performed to accommodate the broad range of transcripts being processed. However, reverse transcription efficiency is influenced by transcript-specific features such as RNA secondary structures[45,46] and unfavorable sequence composition, including high GC content[47,48]. These properties may cause premature enzyme dissociation from the transcribed RNA, generating incomplete cDNAs (i.e., not including the 5′ end of a transcript), potentially impacting ROI information

depending on the position of the dissociation. In addition, reverse transcriptases lack proofreading activity[49,50], which can result in mutations or sequence errors during cDNA synthesis. If the change in sequence consistently occurs at the same position in the template, assignment of nucleotide polymorphisms may be incorrect[50], especially problematic if these sequences belong to ROIs. Finally, the efficiency of the reverse transcription step has been found to be dependent on the choice of reverse transcriptase enzymes[48–54].

Biases in reverse transcription of transcript pools are currently very difficult to overcome, but this step can be omitted by using direct RNA sequencing (DRS)[55,56], in which RNAs are directly sequenced using fluorescent oligonucleotides, or by sequencing with ONT[57]. Additionally, the primer site can be influenced by ligating adapters to RNAs[58]. While these methods circumvent the need for cDNA synthesis, they obtain information in bulk and currently cannot achieve high throughput.

## Template switching
Most reverse transcriptases used in scRNA-seq protocols exhibit two distinct activities commonly referred to as template switching (TS): (1) strand invasion, in which the nascent cDNA dissociates from its original RNA template and anneals to a homologous region on a different transcript, and (2) scRNA-seq template switching (scRNA-seq TS), a term introduced in this work to describe a protocol-specific mechanism driven by the enzyme's terminal transferase activity[59].

Strand invasion can lead to the formation of chimeric cDNAs (e.g., as false fusion transcripts or circular RNAs) or deletions, complicating the accurate detection of alternative splicing events[60–62]. In contrast, scRNA-seq TS involves the addition of untemplated cytosines to the 3′ end of the newly synthesized cDNA when the reverse transcriptase reaches the 5′ end of the RNA[59]. A template switching oligonucleotide (TSO) then anneals to these cytosines and serves as a new template for cDNA extension, enabling capture of full-length transcripts or enrichment for 5′ ends[63–65]. However, this reaction is inherently inefficient[66] as it depends on the enzyme successfully transcribing the entire RNA molecule to its 5′ end. While the template switching efficiency is difficult to quantify, Hughes et al.[66] implemented a randomly primed second-strand synthesis method (called Seq-Well S³) to recover cDNA molecules where scRNA-seq TS had failed. The authors were able to increase gene detection by tenfold compared to samples without recovery through random priming, highlighting the importance of this step.

Both strand invasion and scRNA-seq TS are influenced by the dissociation dynamics of reverse transcriptase, which in turn depend on transcript-specific properties such as sequence composition, secondary structure, and length.

## Generation of shorter cDNAs
Generation of cDNAs between 200 and 600 base pairs (bp) in length is required for scRNA-seq libraries to be compatible with short-read sequencing[67–69]. Two main approaches are used to obtain cDNAs of the appropriate length: tagmentation and random priming. During tagmentation, a transposase simultaneously fragments DNA and appends adapters needed for downstream amplification, whereas random priming involves primers with random sequences binding at different positions along cDNAs. In addition, cDNA size is controlled during library cleanup using size-selection beads, which remove unfavorably sized fragments (including primer dimers) that could bind to flow cells without producing usable data.

Tagmentation introduces several biases[70]. Importantly, only fragments with the correct adapter attached at the correct end of the cDNA are compatible with amplification and sequencing (Fig. 1b). This occurs for 50% of fragments, while the remainder will undergo linear amplification or will not be amplified[70]. Furthermore, transposase insertion sites are not entirely random but exhibit sequence preferences, which can lead to uneven coverage[71–75].

Random priming also introduces several biases, many of which are in common with reverse transcription reactions using random primers. Primers may partially anneal to non-complementary regions, typically with perfect base pairing at the 3′ end but mismatches at the 5′ end. Upon extension, these primers will be incorporated into the second cDNA strand, resulting in cDNAs with altered or artifactually extended 5′ ends[76]. These ends should be removed during data analysis, as they can lead to false-positive mutation calling and biased mapping. Primers may also be sequestered by highly expressed or longer transcripts[48], contributing to the underrepresentation of lowly expressed transcripts in the final library. Finally, structural bias arises when stable RNA secondary structures prevent primer binding, resulting in localized coverage loss.

To be informative and amplifiable in scRNA-seq protocols, cDNAs generated by tagmentation or random priming must have the appropriate size and contain both cell barcode and unique molecular identifier (UMI) sequences. UMIs are unique sequences assigned to each captured or primed transcript and are used to distinguish individual molecules after PCR amplification. Both barcode and UMI sequences are located on barcoded beads adjacent to the captured region. As a result, sequence information is typically limited to regions within ~600 bp of the capture site which, in most cases, is the polyA tail. As the average length of 3′ UTRs in vertebrates is over 800 bp and 100–200 bp for 5′ UTRs[7,77,78], the information obtained with barcoded beads is mostly restricted to these regions, severely impacting retrieval of sequences from coding sequences. This bias is observed for all 3′/5′-based methods and is the most substantial limitation affecting ROI detection.

## PCR biases
PCR amplification is needed to produce sufficient input for HTS. As a general rule, short cDNAs with a balanced GC content composition will be preferentially amplified[79], with regions having high or low GC content exhibiting a biased coverage after sequencing[80–82]. This effect is dependent on the choice of polymerase[83]. Amplification biases related to cDNA length have also been described[80], with efficiencies decreasing for longer cDNAs[84]. Similarly to reverse transcription, PCR conditions are not tailored to amplify specific cDNAs[85], exacerbating the loss of information for unfavorable cDNAs. Transcript representation is also affected by the number of PCR cycles[85], with highly abundant cDNAs being preferentially amplified to the detriment of lowly abundant TOIs. Finally, detection of ROIs in scRNA-seq experiments is also influenced by the fidelity of the polymerase used[86]. Taken together, PCR biases can strongly distort transcript abundance estimates and affect the detection of both TOIs and ROIs, depending on their specific sequences.

## High-throughput sequencing (HTS)
One of the most important parameters to consider when designing RNA-seq experiments is sequencing depth[87] which strongly impacts TOI detection[88,89]. As highly abundant cDNAs will be preferentially sequenced[88], the possibility of detecting lowly expressed transcripts underrepresented in the library increases with higher sequencing depth. Information obtained after HTS sequencing is also influenced by the choice of read length, particularly relevant for ROI detection. Depending on the distance of the ROI to the 3′ end of cDNAs, the sequence may be detected by using longer reads. HTS sequencing also introduces sequencing errors, known to be influenced by GC-rich motifs which often precede these inaccuracies[82,90,91]. The pattern of sequencing errors varies between short-read and long-read sequencing platforms such as PacBio and ONT. Although long-read technologies previously had higher error rates than short-read sequencing, recent advances have greatly improved their accuracy, with current error rates around 99% for ONT, 99.9% for PacBio, and 99.9% for Illumina[92] (the most popular short-read sequencing platform to date[93]). Careful optimization of sequencing depth and read length is thus essential to maximize recovery of low-abundance transcripts and reliably identify ROIs, particularly in complex single-cell datasets.

All of the biases described above impact the output from scRNA-seq experiments, influencing TOI and ROI detection. Several methods have been developed to mitigate these biases in a general, untargeted manner (Box 1). These methods also act at various levels, including sample

## Box 1 | Beyond targeted methods: additional solutions to overcome biases in scRNA-seq experiments

Several solutions have been developed to address the biases described above. These methods do not focus on a given TOI or ROI, but aim to improve the overall information obtained from scRNA-seq experiments. They encompass a wide variety of methods acting at various levels during the scRNA-seq workflow.

*Increasing information on cells of interest during sample preparation:* The main research focus of some scRNA-seq experiments is analyzing rare cell populations or specific cell types. Traditionally, such cells are enriched using FACS sorting, magnetic beads, or specialized isolation protocols. However, defining these populations can be challenging when transcriptomic signatures must be translated into protein markers, especially for transcription factors that often lack suitable antibodies. This is especially the case for FACS sorting, as this technology depends on the available antibodies or reporter strains. To overcome these challenges, recent methods have focused on isolating cells of interest based on the transcripts they express. For example, PERF-seq[148] and EnrichSci[149] use probe-specific detection of transcripts in cell, which are then FACS-sorted for downstream scRNA-seq analysis. By enriching for specific transcript-defined populations, these approaches enhance the recovery and resolution of information from cells of interest.

*Depleting highly abundant RNAs:* Several methods have been developed to deplete highly expressed, non-informative transcripts, mainly rRNAs. This is achieved through specific nucleases[150], by CRISPR/Cas9 degradation with guide RNAs against the transcripts to deplete[38,151,152,153], by digestion with RNAseH[23] or by using primers to block the extension or reverse transcription of selected transcripts during PCR amplification[33,154]. All of these methods lead to similar results: depletion of highly expressed, non-informative transcripts increases the detection of lowly expressed RNAs in scRNA-seq experiments. For example, through scCLEAN[153] the authors were able to more than double the percentage of non-targeted UMIs (from 39% to 92%). Depleting highly abundant RNAs frees additional oligonucleotides on barcoded beads for mRNA capture (reducing dT capture/priming bias). It also benefits lowly expressed transcripts by increasing the available read depth and preventing their underrepresentation during PCR amplification.

*Using a transcript polyadenylation step to profile non-polyadenylated transcripts:* Non-polyadenylated transcripts are missed in scRNA-seq experiments based on dT capture/priming. These transcripts are highly relevant for many biological questions, including detection of viral, bacterial and non-coding RNAs (such as microRNAs, snoRNAs and some lncRNAs). To overcome this challenge and obtain this information in combination with polyadenylated RNAs, several methods have included transcript polyadenylation. These include VASA-seq[23], where transcripts are first fragmented after which each fragment gets a polyA tail, Smart-seq-total[155], where all transcripts are polyadenylated (followed by depletion of rRNA transcripts) and SUPeR-seq[34], SMARTer[35] and MATQ-seq[22], with reverse transcription based on random primers carrying an adaptor.

*Overcoming PCR biases:* PCR bias in scRNA-seq experiments is mitigated by using UMIs to control for over-amplification of cDNAs. Several additional methods have been developed for bulk RNA-seq libraries. Kozarewa et al.[156] designed a protocol relying on ligation of Illumina sequencing adapters directly to first-strand cDNAs. Compared to standard bulk RNA-seq protocols using PCR amplification, lower GC coverage bias and duplicate sequences were observed. An alternative method, smsDGE[157] relies on hybridization of first-strand cDNAs to flow cells, followed by direct sequencing using fluorescent oligonucleotides. Mamanova et al.[158] developed FRT-seq, which relies on reverse transcription of RNAs directly on flow cells, allowing sequencing of first cDNA strands. While these methods have not been extented to scRNA-seq experiments, linear amplification through in vitro transcription using a T7 DNA-based RNA polymerase has emerged as an alternative for minimizing the number of PCR cycles. Methods using this step include MARS-seq[159], CEL-seq[8] and LAST-seq[160].

---

generation, capture on barcoded beads and PCR. However, they cannot specifically enrich for information on TOI and ROIs, which may still missed.

## Targeted scRNA-seq strategies using short-read sequencing

The loss of information on TOIs and ROIs introduced by the biases described above is addressed by targeted methods acting at different steps in the scRNA-seq protocol. We have subdivided these methods into five different classes based on their targeted enrichment strategy (Fig. 2a): (1) targeted capture, (2) targeted priming, (3) targeted amplification, (4) dual targeted PCR and (5) probe hybridization. Each category of targeted methods addresses a certain subset of biases (Fig. 2a) and is associated with both advantages and inherent limitations, outlined in each section of the text. A timeline of the development of targeted scRNA-seq methods is outlined in Fig. 2b, while a schematic representation of their protocols is shown in Fig. 3. An overview of their key characteristics, including number of targets, number of cells and detection of the standard transcriptome is provided in Table 2.

### Category 1: Targeted capture

Targeted capture is based on hybridization of cellular RNAs to barcoded beads modified with oligonucleotides complementary to internal regions of targets of interest (TOIs). These capture sequences physically bind TOIs from the diverse RNA pool, enriching them prior to reverse transcription and enhancing their detection while maintaining coverage of the standard cellular transcriptome (the transcriptome obtained in the non-targeted version of the method). Targeted capture methods tackle five biases (Table 1, Fig. 2a): dT capture/priming, reverse transcription, 3′/5′ bias, read depth and read length. The dT capture/priming step is addressed with the capture sequence on the beads by influencing the position from which reverse transcription initiates. By initiating reverse transcription at internal regions, rather than at the transcript ends, information which would be limited to the 3′ or 5′ end of transcripts can be shifted, increasing ROI detection and decreasing the read length needed to reach the ROI(s).

Three examples of methods using targeted capture are DART-seq[94], RoCKseq[95] and direct capture Perturb-seq[96]. DART-seq and RoCK-seq are both based on solid barcoded beads. In DART-seq, a subset of the poly(dT) oligonucleotides on Drop-seq beads is modified with capture sequences, allowing simultaneous enrichment of TOIs and detection of the standard transcriptome. In contrast, RoCK-seq modifies only the template-switch oligonucleotides (TSOs) on BD Rhapsody beads, leaving all poly(dT) oligonucleotides available for standard transcriptome capture. The two methods use different strategies for bead modification: DART-seq uses a ligase to attach a partially double-stranded oligonucleotide, whereas RoCK-seq employs a polymerase-based approach using a single-stranded splint oligonucleotide as a template. These differences influence the properties of the two methods. DART-seq achieves a variable modification rate ranging between 20 and 40%, while RoCKseq achieves modification rates near completion, which can be titrated, multiplexed or used in combination with different assays. As DART-seq uses the same dT oligonucleotides for both

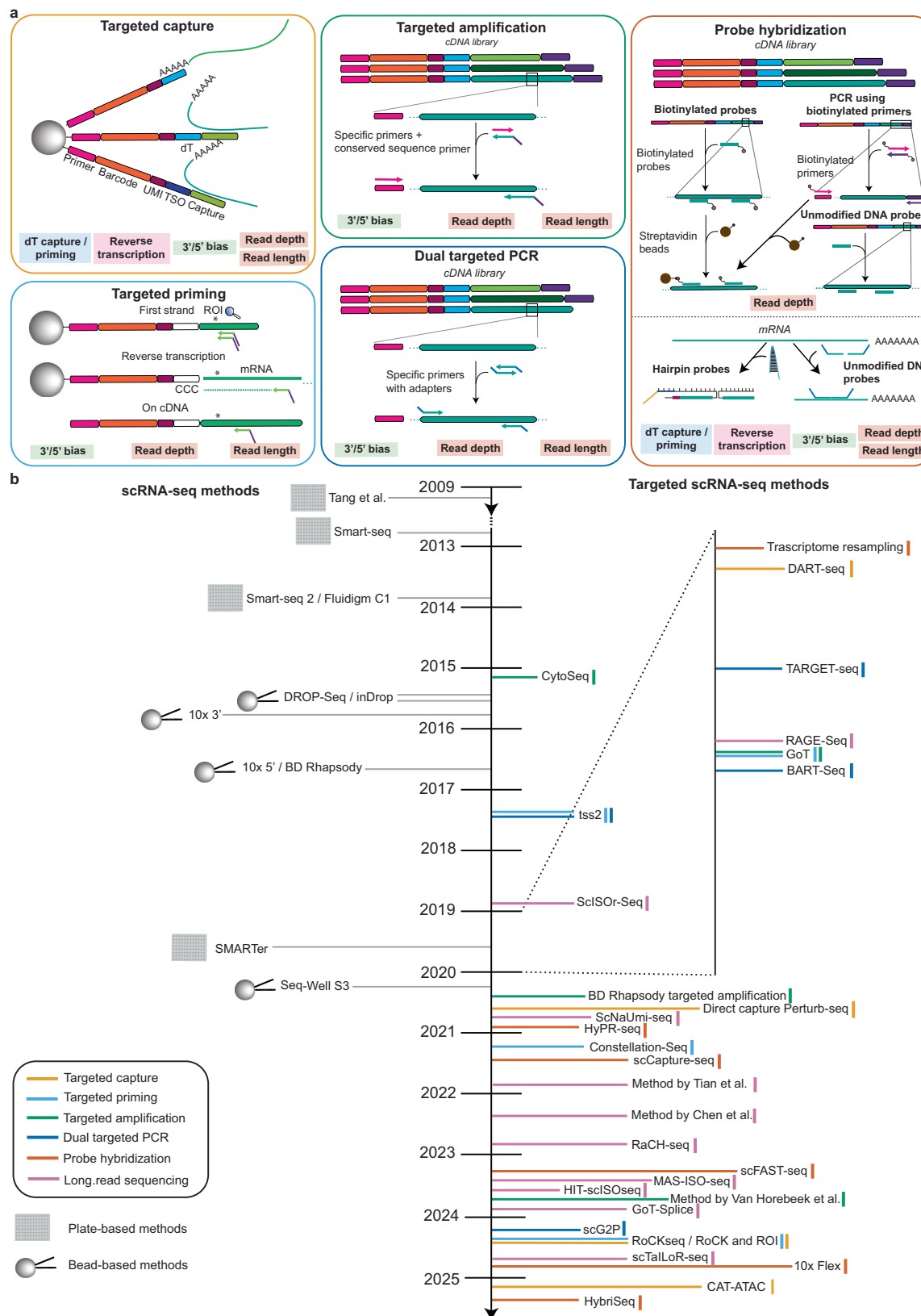

**Fig. 2 | Categorization of five targeting strategy in scRNA-seq experiments. a** Five categories of targeted scRNA-seq methods using short-read sequencing. The biases which they address are listed below. All of the methods except for targeted capture can be directly applied in the context of current scRNA-seq technologies using either barcoded beads or dT primers for mRNA capture. **b** Timeline representing the targeted scRNA-seq methods and the release of the platforms which they are based on. For all methods the release date of the preprint or of publishing of the paper was used. Methods shown in the figure include CytoSeq[15], tss2[99], ScISOr-Seq[119],

Transcriptome resampling[106], DART-seq[94], TARGET-seq[104], RAGE-Seq[118], GoT[98], BART-Seq[103], BD Rhapsody targeted amplification[100], Direct capture Perturb-seq[96], ScNaUmi-seq[120], HyPR-seq[110], Constellation-Seq[97], scCapture-seq[107], Method by Chen et al.[126], RaCH-seq[122], scFAST-seq[33], MAS-ISO-seq[124], HIT-scISOseq[123], Method by Van Horebeek et al,[102] GoT-Splice[101], scG2P[105], RoCKseq / RoCK and ROI[95], scTaILoR-seq[125], 10x Flex[109], CAT-ATAC[145], HybriSeq[108], Method by Tian et al.[121]

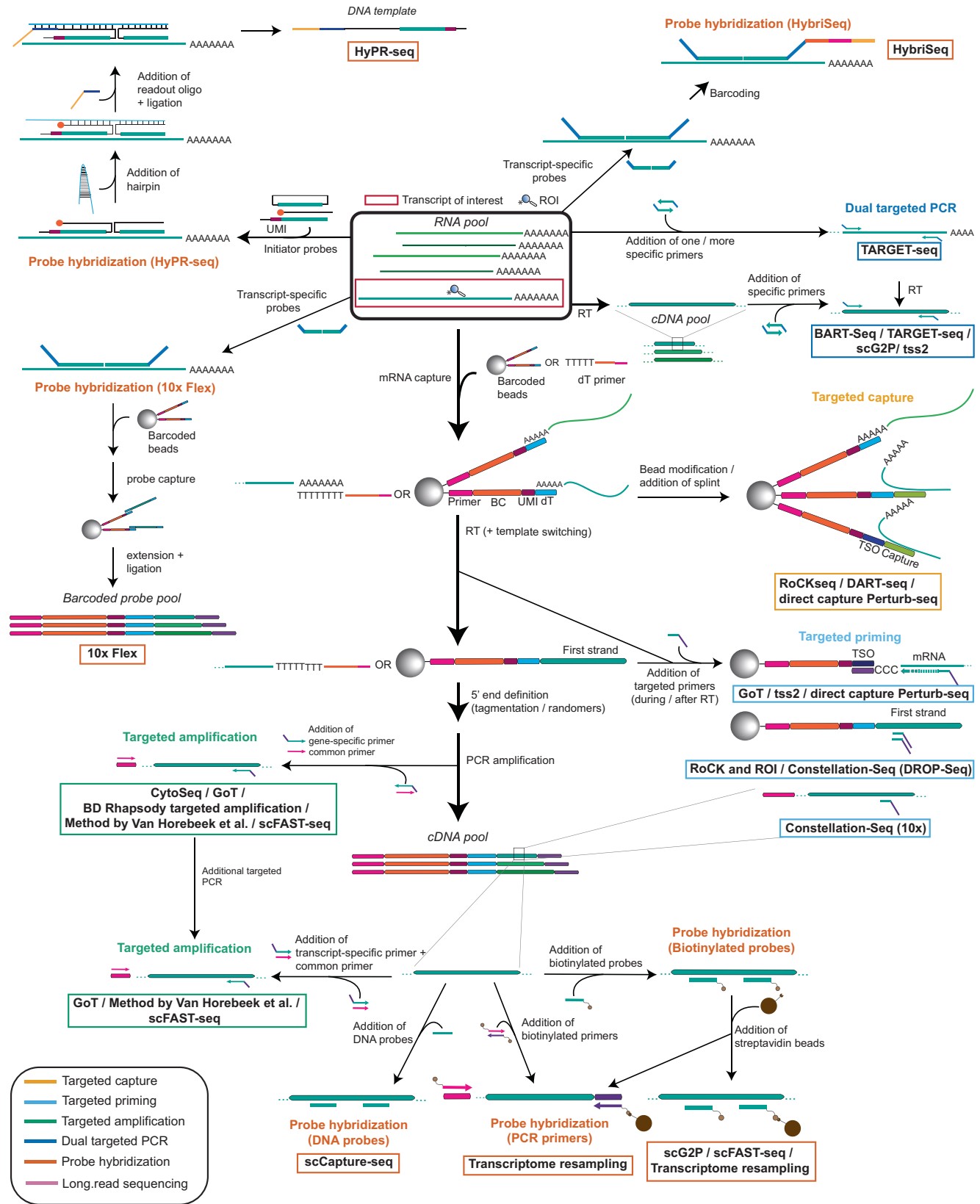

**Fig. 3 | Summary of targeted scRNA-seq methods.** RT: reverse transcription. Methods belonging to the five categories are colored accordingly. Arrows in bold indicate the main steps of the scRNA-seq protocol and are subdivided into dT priming and capture, while lighter arrows indicate additional steps which are needed for specific targeting towards TOI and ROI detection. Methods shown in the figure include CytoSeq[15], tss2[99], Transcriptome resampling[106], DART-seq[94], TARGET-seq[104], GoT[98], BART-Seq[103], BD Rhapsody targeted amplification[100], Direct capture Perturb-seq[96], HyPR-seq[110], Constellation-Seq[97], scCapture-seq[107], scFAST-seq[33], Method by Van Horebeek et al.[102], scG2P[105], RoCKseq / RoCK and ROI[95], 10x Flex[109], HybriSeq[108].

## Table 2 | Main features of targeted scRNA-seq methods

| Method | Platform they are based on | Throughput | Type of targeting | Show targeting of TOIs | Show targeting of ROIs | # targets described | Standard transcriptome | Show targeting of TOIs | Show targeting of ROIs |
|---|---|---|---|---|---|---|---|---|---|
| CytoSeq | Custom | medium | Targeted amplification | yes | no | 111 | no | yes | no |
| tss2 | Smart-seq 2/ Fludigm C1 | low | Targeted priming / dual targeted PCR | yes | yes | 1 | yes | yes | yes |
| BART-Seq | Custom | low | Dual targeted PCR | yes | yes | 15 | no | yes | yes |
| DART-seq | DROP-Seq | high | Targeted capture | yes | yes | 2 | yes | yes | yes |
| GoT | 5'/V(D)J 10x Chromium and 3' 10x Chromium | high | Targeted priming at reverse transcription level / targeted amplification | yes | yes | 5 | yes | yes | yes |
| TARGET-seq | Custom | medium | Dual targeted PCR | no | yes | 12 | yes | no | yes |
| Transcriptome resampling | 3' 10x Chromium | low | Probe hybridization (biotin) | yes | no | NA | yes | yes | no |
| BD Rhapsody targeted amplification | BD Rhapsody | high | Targeted amplification | yes | no | 492 | no | yes | no |
| Direct capture Perturb-seq | 5'/V(D)J 10x Chromium and 3' 10x Chromium | high | Targeted capture / targeted priming | yes | yes | 1 | yes | yes | yes |
| HyPR-seq | Custom | medium | Probe hybridization (hairpin probes) | yes | yes | 179 | no | yes | yes |
| Constellation-Seq | DROP-Seq/3' 10x Chromium | high | Targeted priming on amplified cDNAs or first strands | yes | no | 127 | no | yes | no |
| scCapture-seq | Smart-seq, Fluidigm C1 and Smarter-seq | low | Probe hybridization (DNA probes) | yes | no | 972 | yes | yes | no |
| scFAST-seq | Custom | medium | Targeted amplification / probe hybridization (biotin) | yes | yes | 2 | yes | yes | yes |
| Method by Van Horebeek et al. | 3' 10x Chromium | high | Targeted amplification | yes | yes | 1 | yes | yes | yes |
| RoCK and ROI | BD Rhapsody | high | Targeted capture / targeted priming | yes | yes | 8 | yes | yes | yes |
| scG2P | Custom | medium | Dual targeted PCR / probe hybridization (biotin) | yes | yes | 56 | no | yes | yes |
| 10x Flex | 3' 10x Chromium | high | Probe hybridization | yes | no | 19,000 | (no)* | yes | no |
| HybriSeq | Custom | medium | Probe hybridization | yes | no | 95 | no | yes | no |

The throughput relates to the number of cells. Low throughput is in the range of a few hundreds of cells. Medium throughput is up to 10,000 cells, and high throughput is upwards of 10,000 cells. Methods shown in the table include CytoSeq[15], tss2[99], Transcriptome resampling[106], DART-seq[104], TARGET-seq[94], GoT[88], BART-Seq[103], BD Rhapsody targeted amplification[100], Direct capture Perturb-seq[96], HyPR-seq[110], Constellation-Seq[97], scCapture-seq[33], scFAST-seq[23], Method by Van Horebeek et al.[102], scG2P[105], RoCKseq/ RoCK and ROI[85], 10x Flex[109], HybriSeq[108]. *10x Flex protocol: although over 18'000 genes are targeted through probes, the transcriptome information depends on the choice and position of probes.

targeted and whole-transcriptome capture, its multiplexing capacity is limited. However, in both cases targeted capture was shown to increase the amount of information compared to samples using unmodified beads (430-fold increase for DART-seq and 379.7 increase for RoCKseq across different regions of interest). Direct capture Perturb-seq is based on the $10 \times 3'$ Chromium platform. Similar to DART-seq, it relies on addition of a partially double-stranded oligonucleotide. Since 10× beads are not solid, modification cannot be done in advance. Reagents are added directly to the reverse transcription mix, and the modification rate cannot be assessed prior to the scRNA-seq run. Direct capture Perturb-seq modifies both cs1 or cs2 oligonucleotides on 10× beads, leaving the dT oligonucleotides free for polyA capture. While the method was designed for guide RNAs in CRISPR/Cas9 perturbation assays, it can also be expanded to other targets.

A key advantage of targeted capture methods is that they address the earliest step of library generation, ensuring TOI incorporation into the sequencing library. Regions unfavorable for reverse transcription (e.g., GC-rich sequences) can be bypassed, and both polyadenylated and non-polyadenylated transcripts can be targeted. Limitations include the potential for off-target capture at sequence motifs resembling the designed capture sequence. This issue is exacerbated by the hybridization conditions of RNAs on barcoded beads, which occurs at low temperatures (for the 10× and BD Rhapsody platform room temperature with ice-cold reagents), below the melting temperatures of the capture sequences. Additionally, hybridization at TOIs may be inefficient as the targeted sequences may be buried inside RNA secondary structures.

Despite these caveats, targeted capture remains a valuable approach, offering efficient enrichment for TOIs while preserving comprehensive transcriptome coverage. As all of these methods are built on widely used high-throughput commercial platforms, they can be easily integrated into existing workflows with minimal adjustments. The optimal choice of targeted capture method ultimately depends on the available platform and experimental resources.

## Category 2: Targeted priming

Targeted priming methods use primers which bind to either transcripts or cDNAs to enrich for the information on ROIs and/or TOIs, followed by reverse transcription or second strand synthesis, respectively. By defining the 5′ ends of cDNAs and shifting the information to regions within TOIs, they mitigate the 3′/5′ bias and require a decreased read length to reach ROIs.

Targeted priming methods are distinguishable based on the step in which the targeting occurs: first strand cDNAs for RoCK and ROI[95] and Constellation-Seq[97], reverse transcription for GoT[98], direct capture Perturb-seq[96] and tss2[99] and amplified cDNAs for Constellation-Seq. The Constellation-Seq authors applied their method to both 3′ 10× Chromium and DROP-Seq technologies (Fig. 3). For DROP-Seq libraries, specific primers are added after first-strand cDNA synthesis, and the authors report an increase of UMI counts of 2.7 times compared to the untargeted condition. Targeted priming on 3′ 10× Chromium libraries is directly applied to PCR-amplified cDNAs by using a transcript-specific primer. Compared to standard 10×, the authors calculated a 22-fold greater sensitivity through targeting. The authors of direct capture Perturb-seq described TOI enrichment on both 3′-based 10× Chromium libraries (through targeted capture) and 5′/V(D)J 10× Chromium (through targeted priming). tss2[99] is the only out of the targeted priming methods which is not based on 3′/5′ methods but on Smart-seq technology and thus associated with lower cellular throughput. However, this method is highly sensitive, with the authors increasing ROI detection from 25% to 100%.

Advantages of targeted priming include ease of implementation and flexibility, as the specific primers can be spiked into a pre-existing reaction. However, these methods strongly depend on the sequence of the transcript itself, which may not be favorable for binding of primers. Priming to other transcripts may also occur, leading to the detection of off-targets[97]. Additionally, targeted priming at the level of reverse transcription cannot be applied when using barcoded beads and 3′-based library generation

protocols, as the reverse transcription reaction initiates directly on the beads to preserve critical barcode and UMI information. Applying targeted priming to already amplified cDNAs may introduce additional biases, as PCR amplification can already skew transcript representation. Consequently, targeted priming is most applicable at the level of reverse transcription (in the context of 5′ or V(D)J 10× Chromium workflows) or during first-strand cDNA synthesis.

## Category 3: Targeted amplification

Targeted amplification methods use PCR to enrich for TOIs by combining a universal primer sequence with a transcript-specific primer, thereby retaining information on the cellular barcode and UMI of each TOI (Fig. 2a). The universal primer sequence is added to all cDNAs after reverse transcription and is used for PCR amplification; it is located adjacent to the barcode. The transcript-specific primer has the same adapter appended during tagmentation or random priming, allowing subsequent amplification of the cDNAs during standard library generation. Targeted amplification methods tackle three biases: 3′/5′ bias, read length and sequencing depth. These methods can recover information on multiple TOIs, but the standard transcriptome may not always be simultaneously profiled. This is the case for CytoSeq[15], which uses a set of primers to amplify transcript panels after first-strand synthesis on barcoded beads (Fig. 3). The same principle is used by the BD Rhapsody targeted amplification system[100] which scaled up the number of targets to several hundred. Other scRNA-seq methods using targeted amplification, which also profile the standard transcriptome, include GoT[98] and its derivative GoT-Splice[101]. As described above, in GoT a specific primer is added during reverse transcription while generating 5′/V(D)J 10× Chromium libraries. Libraries are then split into two: one part of the sample undergoes standard 10× Chromium indexing with PCR and fragmentation, while the second part (around 10%) undergoes a round of targeted amplification. Through targeted amplification and priming, the GoT-seq authors were able to show that for their mutation of interest in the *CALR* transcript, genotyping data were available for 88.7% of cells, compared to only 1.4% of cells in the untargeted condition. The same protocol (without targeted priming) is followed by GoT-Splice. Targeted amplification was also used by Van Horebeek et al.[102], who were able to increase multiple targets from below 1% to over 80%. Similar to GoT, this method is based on using part of the 3′ 10× Chromium output after first-strand synthesis to perform two rounds of targeted amplification, thus enriching for TOIs.

Targeted amplification is cost-effective, flexible, and compatible with existing cDNA libraries. Additionally, compared to targeted capture, annealing of primers occurs close to their melting temperature, with lower chances of off-target binding. However, it introduces PCR-related biases, and targeting efficiency depends on cDNA sequence and reaction conditions. Multiplexed detection can be uneven, as uniform PCR conditions may favor some targets over others. The BD Rhapsody targeted amplification authors in fact showed that 25% of the amplified transcripts actually had a lower detection level compared to the standard protocol, even after multiple PCR rounds[100]. Importantly, if the initial RNA capture or reverse transcription was inefficient for a given transcript, further PCR cannot recover the information. Despite these limitations, targeted amplification remains a suitable approach for amplifying selected targets from pre-generated cDNA libraries, such as when retrieving additional information after sequencing. Nonetheless, this strategy is not recommended for large panels (e.g., >10 targets), as PCR conditions cannot be optimized for each primer set.

## Category 4: Dual targeted PCR

Dual targeted PCR methods use transcript-specific forward and reverse primers for TOI amplification. In contrast to the previously described targeting categories, these methods do not retain information on the cell barcode and UMI but can instead introduce cellular indices attached to transcript-specific primers. This strategy is used by BART-Seq[103], a plate-based technology described for both bulk and scRNA-seq. The TARGET-seq[104] plate-based, multiomic method uses a combination of targeting

genomic DNA, as well as cDNA, for the detection of mutated genes and their expression in cancer cells. TARGET-seq is based on two rounds of targeted PCR, one directly on mRNAs while they are being reverse-transcribed and one on previously generated cDNAs. Similar to TARGET-seq, scG2P[105] is a plate-based method combining sequencing of genomic DNA with targeted PCR, which occurs after reverse transcription using primers harboring cell-specific barcodes. Finally, tss2, which as outlined above uses targeted priming at the reverse transcription level, further enhances the information on ROIs usings targeted PCR by appending specific adapters used for subsequent PCR amplification.

As both targeted amplification and dual-targeted PCR methods are based on PCR amplification, they share common advantages and disadvantages. However, methods using dual-targeted PCR do not retain cellular barcode information and require the addition of cellular indices to the primers used for amplification. This implies that UMI-based correction for PCR amplification cannot be performed. Additionally, all dual-targeted PCR methods rely on custom set ups and, except for scG2P, are limited in the number of profiled cells and thus not compatible with high-throughput technologies. Given these factors, dual-targeted PCR methods currently appear less scalable and less broadly applicable than other targeting strategies.

## Category 5: Probe hybridization

Targeted scRNA-seq approaches using probe hybridization to enhance TOI detection are the most diverse category of strategies. There are three main types of probe hybridization methods based on the type of enrichment: biotin-based enrichment, unmodified DNA probes or hairpin probe hybridization.

In approaches using biotinylated probes, the probes first hybridize to their target cDNAs, which are then selectively recovered using streptavidin-coated beads that bind to the biotin moiety (Fig. 3). This strategy is used by Transcriptome resampling[106]. The goal of the study was to obtain information on a specific subset of cells in a dataset generated with 3′ 10× Chromium chemistry. Biotinylated probes recognizing the respective cell barcodes were used to re-sequence them at a higher depth. The authors also used biotin probes as well as PCR primers conjugated with biotin to isolate TOIs from libraries. Through their probe hybridization strategy, the authors were able to increase the number of UMIs by up to 2.85-fold compared to the non-targeted cells. Additionally, the percentage of positive cells increased from 59.7 to 100% for a given TOI. Another scRNA-seq method using biotinylated probes for the detection of TOIs is scFAST-seq[33]. It relies on either two rounds of targeted amplification, or enrichment of transcripts via biotinylated probes followed by PCR amplification.

The second type of probe hybridization method uses linear DNA probes without modification, as exemplified by scCapture-seq[107]. Following hybridization to the target cDNAs, the resulting double-stranded DNA complexes are isolated and subsequently PCR-amplified to increase their abundance. Through this technology, the authors showed an increase in reads mapped to transcription factors (their TOIs) from 2.2 to 78.3% (36-fold), with target gene expression increasing by 150-fold on average. In contrast to scCapture-seq, HybriSeq[108] is a plate-based method using DNA probes which recognize two adjacent regions on RNAs. The bound probes are then ligated, followed by split-pool barcoding to tag each cell with a specific barcode, PCR amplification and sequencing. A similar strategy is used by the 10× Flex protocol[109], in which probes also carrying a specific index sequence and adapters, are hybridized to mRNAs of interest. Compared to the HybriSeq protocol, barcodes are added through 10x beads, as the probes contain an adapter which is complementary to a sequence on the barcoded beads and can thus be subsequently captured. As of fall 2025, the 10× Flex probe library includes probes against more than 18,000 genes for mouse and 19,000 for human. While this is still a targeted method, the resolution is extremely high and close to the information obtained in the standard transcriptome. Additionally, by having multiple probes per gene this technology is less dependent on efficiency of probe design (which might for example be impacted by mutations in transcripts). Both the 10× Flex

protocol and HybriSeq can be applied to fixed and permeabilized tissues, where RNA quality is often lower compared to fresh tissue samples.

The third class of probe hybridization methods employs hairpin probes, as in HyPR-seq[110]. This approach targets RNAs directly using two initiator probes that hybridize to adjacent sequences on the transcript of interest (TOI). After addition of further DNA oligonucleotides (plate-based), including a hairpin probe, cDNA constructs containing the sequence of the targeted RNAs, adapters, and UMIs are generated. The final cDNA constructs are then annealed to barcoded microbeads, thus retaining the information at the single-cell level. By comparing their method to existing 10× Chromium data, the authors calculated that using 10× they would need over 1,000,000 cells at 20,000 reads per cell to detect 25% of expression changes in 90% of genes expressed above 1 transcript per million. In contrast, their method achieved the same power with only about 25,000 cells at 5000 reads per cell.

Similar to targeted amplification and dual-targeted PCR, one of the main advantages of probe hybridization methods acting on cDNA is their flexibility and compatibility with different platforms. A key challenge across all probe hybridization methods lies in designing probes with sufficient target specificity, as sequence variations in RNA or cDNA can impair probe binding. For methods applied to cDNA, enrichment of TOIs is further constrained by earlier steps in library preparation and overall library complexity.

In contrast, probe hybridization methods targeting RNA act at the earliest stage of the scRNA-seq workflow. Historically, these approaches were labor-intensive, required custom setups, and thus have not been widely adopted. The 10× Flex protocol represents a major exception, enabling scalable, high-throughput targeting of thousands of transcripts on the 10× Chromium platform. However, it is not suited for ROI detection, as it sequences ligated probes rather than the RNAs themselves. Moreover, probes are currently available only for human and mouse samples. Because tens of thousands of genes are targeted simultaneously, this approach still entails some information loss for specific TOIs. Finally, non-templated ligation events may occur, contributing to background noise[111]. Despite these limitations, the 10× Flex protocol introduced a breakthrough in the use of fixed and permealized tissues for scRNA-seq experiments.

While targeted methods tackle most biases described in the section above, none of them comprehensively addresses PCR biases (Fig. 2a). This would require changing the PCR conditions to specifically fit TOIs, which is not possible when multiple transcripts with different properties are amplified in the same reaction and under the same conditions. Such an approach however is a valuable solution and intrinsically very sensitive for single targets of interest.

## Targeted sequencing combined with long-read technologies

The length of a single ROI which can be detected with the methods described above is limited to ~600 bp or restricted to a low number of specific, but short transcript regions. For example, studying novel isoforms in a long transcript using short-read targeted scRNA-seq methods requires targeted capture or priming using several primers or PCR reactions in order to recover information from multiple ROIs[94]. To overcome these limitations, long-read sequencing has been recently combined with 3′/5′-based methods, obtaining full-length transcript profiles[24,112–116]. Information obtained through long-read sequencing and 3′/5′-based methods is connected through detection of the barcode sequence.

Combining long-read sequencing technologies with 3′/5′-end-based short-read sequencing platforms presents two major challenges (Fig. 4). Several biases limit recovery of information from the full transcript length (internal capture/priming and dissociation of the reverse transcriptase enzyme). Additionally, similar to short-read platforms and as long-read sequencing experiments typically yield lower read depth than short-read sequencing[117], TOIs need to be enriched for in the library prior to sequencing. Both challenges have been addressed using targeted enrichment

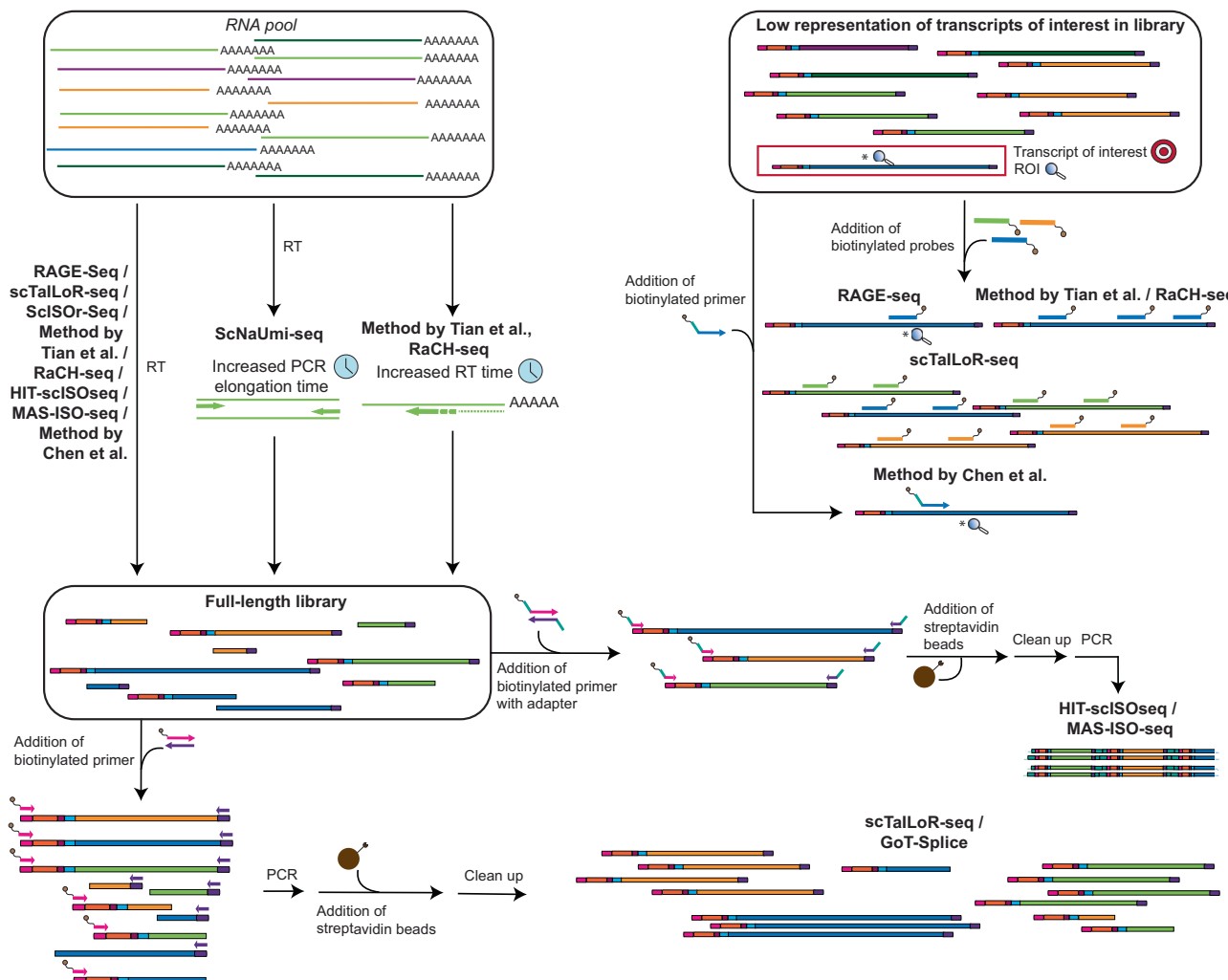

**Fig. 4 | Summary of technologies coupling long-read sequencing with 3′/5′ based library generation.** The full-length library also shows fragmented transcripts which were generated by biases during early library generation steps, including reverse transcription. Methods shown in the figure include ScISOr-Seq[119], RAGE-Seq[118], ScNaUmi-seq[120], Method by Chen et al.[126], RaCH-seq[122], MAS-ISO-seq[124], HIT-scISOseq[123], GoT-Splice[101], scTaILoR-seq[125], Method by Tian et al.[121].

strategies. An overview of the key features and implementation stages of the methods discussed below is provided in Fig. 4.

## Application of targeting strategies to long-read protocols
Early long-read sequencing methods adapted to 3′/5′-based technologies lacked effective strategies for enriching full-length cDNAs with barcode sequences. Examples include RAGE-Seq[118] and ScISOr-Seq[119]. In RAGE-Seq, only 18.7% of reads contained barcode information, while in ScISOr-Seq, 58% of reads carried barcodes, out of 61.6% that exhibited a detectable oligo(dT) tail marking the 3′ end of the cDNA.

To improve the recovery of full-length, barcode-containing transcripts, several protocol modifications were introduced. ScNaUmi-seq[120] increased PCR elongation time from 1 to 3 min, while Tian et al.[121] and RaCH-seq[122] extended the reverse transcription step to 2 h. Other approaches, including HIT-scISOseq[123], MAS-ISO-seq[124], and GoT-Splice[101], used targeted priming with a biotinylated universal primer to selectively enrich cDNA fragments that contain both the cell barcode and UMI. Similarly, scTaILoR-seq[125] employed biotinylated PCR primers to enhance the recovery of barcode-containing sequences.

Beyond barcode enrichment, several long-read methods have leveraged biotinylated probes or primers to target TOIs (Fig. 4). RAGE-Seq uses biotinylated capture probes directed at specific transcript regions, while ScISOr-Seq and RaCH-seq extend this strategy across the full cDNA length. Chen et al.[126] implement targeted priming using a biotinylated primer positioned adjacent to a fusion breakpoint of interest in 10× Chromium libraries. In scTaILoR-seq, biotinylated probes are used to enrich for over 1000 specific genes, boosting TOI recovery from 5% to 95%. Notably, these enrichment strategies do not require precise targeting of ROIs, as long-read sequencing enables recovery of internal transcript regions.

Despite these advancements, long-read protocols adapted to 3′/5′-based technologies require additional PCR amplification to increase input material, potentially introducing further bias. Additionally, these methods are all built upon libraries generated through oligo(dT)-based capture or priming, template switching, and reverse transcription. As detailed in Table 1, these three steps introduce sequence-dependent biases, including the generation of fragmented constructs, which may affect transcript coverage and interpretation.

## Targeted methods in spatial biology
Targeted technologies are essential in the field of spatial biology, which has emerged as a standard technology to obtain information on RNAs expressed by cells in combination with their spatial position. The field has been rapidly evolving from multi-cell to single-cell resolution and to broad transcript

coverage[127]. Methods can be largely classified into two categories: imaging-based and sequencing-based[128,129] technologies.

### Image-based technologies

Image-based spatial transcriptomic technologies ultimately rely on microscopy-based readouts to obtain spatial information on RNAs. Two types of probe-based, targeted approaches are prevalent in the field (reviewed in Tian et al.[127]). Early methods, such as seqFISH[130] and MERFISH[131] were based on in situ fluorescent hybridization using FISH probes complementary to RNAs of interest[132]. In the second strategy, RNA molecules or in-situ reverse transcribed cDNAs are hybridized with specific probes which are then amplified using rolling circle amplification and visualized with fluorescent probes[127]. As both types of image-based technologies use targeted probe hybridization methods, they suffer from the same biases described above, in particular probe design.

### Sequencing-based technologies

Many sequencing-based spatial transcriptomic platforms, such as the 10× Visium technology[133], rely on capture of RNAs on spatially indexed surfaces covered with DNA oligonucleotides with a barcode specific to a given location on the slide[127]. Alternative solutions to introduce a spatially-resolved barcode are Slide-seq[134] and Stereo-seq[135]. As these methods rely on dT-based capture, information is restricted to the UTRs of transcripts. After RNA capture, the next steps follow similar workflows as the 3′/5′-based methods, with reverse transcription, PCR amplification and sequencing. Targeted amplification and probe hybridization methods can be applied to these libraries to enrich for TOI information, while retaining the spatially-resolved information. This was for example implemented by McKellar et al.[136], who used biotinylated probes to enrich for viral transcripts. Similarly, Engblom et al.[137], used biotinylated probes to enrich for BCR-TCR transcripts, retaining spatial information. B and T cell repertoires were also enriched for by Sudmeier et al.[138], who used targeted amplification with a transcript-specific reverse primer. A second strategy used in sequencing-based spatial transcriptomic methods is probe hybridization to RNAs, followed by sequencing of the bound probes[139]. This strategy is used by the 10× Xenium technology, which is also the basis for the 10× Flex protocol. In this case, the information is not limited to the UTRs of transcripts, but internal ROIs cannot be accessed as the probes themselves are sequenced and not the RNAs themselves.

These developments illustrate how targeted technologies in scRNA-seq and spatial transcriptomics have co-evolved, with targeting strategies transferable between both domains.

## Biological applications of targeted scRNA-seq methods

Targeted scRNA-seq methods have been applied across a wide range of contexts, from detecting non-polyadenylated transcripts to studying cancer mutations, alternative splicing and capturing information on guide RNAs during pooling screens.

### Non-polyadenylated transcripts

As viral and prokaryotic RNAs lack poly(A) tails, they are often invisible in conventional 3′/5′-based scRNA-seq. Targeted strategies resolve this gap. For example, DART-seq enriched reads from reovirus-infected cells by 430-fold compared to standard DROP-seq technologies by directing sequencing towards an internal viral region, enabling mutation-level resolution of viral heterogeneity. Focusing on enhancing detection of bacterial transcripts, INVADE-seq[140] combined targeted priming and PCR to amplify bacterial 16S rRNA alongside host cell transcriptomes in patient tumor samples. This issue is particularly relevant as bacterial cells are estimated to contain more than 100 times less RNA compared to eukaryotic cells, and transcriptional turnover is much faster compared to eukaryotes[141]. The dual profiling revealed that bacteria in the tumor microenvironment are primarily localized within myeloid cells and can modulate host immune responses, with potential implications for immunotherapy outcomes.

### Cancer mutations and fusion events

Another major application of targeted scRNA-seq technologies is cancer research, where mutations often reside in internal transcript regions missed by 3′/5′-based methods. Targeted methods are thus fundamental to not only focus on these mutations, but also to answer relevant questions such as which cell types harbor the mutation and the changes in the cellular transcriptome they introduce. GoT-seq, for instance, enabled mutation detection in CALR and XBP1, linking them to transcriptional changes in specific cell types and highlighting IRE1 signaling as a potential therapeutic target. The focus of the study conducted by Van Horeebek et al. and scFAST-seq was also the detection of disease-causing mutations and their assignment to specific cell types. Of particular interest in many cancer types are fusion breakpoints, as these are often disease-causing mutations. These include the breakpoint of the BCR::ABL1 gene, which is found at more than 3 kb away from the 3′ end of the transcript. Given its relevance in the context of leukemia, several targeted methods focused on directing their information to this ROI. These methods include tss2 and other methods such as scFAST-seq, RoCK and ROI and Nilsson et al.[142], who adapted a version of GoT-seq for the detection of BCR::ABL1.

### Alternative splicing

Targeted methods also make it possible to study alternative splicing with single-cell resolution. These events exclusively occur within internal transcript regions and thus cannot be analyzed with standard 3′/5′-based technologies. scTaILoR-seq, for example, examined over 2200 targeted transcripts to characterize isoform diversity in ovarian cancer, revealing differential isoform usage between cell types. As shown by RoCK and ROI, alternative splicing can also be analyzed with short-read based methods. The authors of this method directed reads to exon–exon junctions of a selected transcript (Pdgfrα), providing a framework for studying splicing variation in defined cell types across TOI.

### Rare cell types and markers genes

Targeted enrichment can also improve detection of rare cell populations and lowly expressed genes, which often serve as cell-type-specific markers. Transcriptome Resampling recovered more information from megakaryocytes, which represent only ~0.1% of bone marrow samples, by resequencing reads from selected barcodes. Constellation-seq increased marker recovery in PBMCs, expanding dendritic cell representation from 51 to 127 cells. Likewise, scCapture-seq enriched transcription factor transcripts, which are notoriously under-detected in scRNA-seq, enabling finer resolution of neuronal subpopulations and increasing the number of differentially expressed genes distinguishing annotated clusters from 129 to 155.

### Capture of guide RNAs

Capturing guide RNAs is essential in pooled CRISPR screens, as it enables linking cellular transcriptome profiles to the corresponding perturbations. Perturb-seq[143] is a widely used approach in which guide constructs are engineered with a poly(A) tail, allowing capture on barcoded beads. Guide recovery is then enhanced through targeted amplification, which associates each perturbation with transcriptomic changes. To increase sensitivity for genes of interest that may be affected by perturbations, TAP-seq[144] introduced targeted amplification of both guide RNAs and selected transcripts. Targeted capture has also been applied specifically to enrich guide RNAs, as in direct-capture Perturb-seq[96] and CAT-ATAC[145]. This last method is a multiomic technology integrating information on guide RNAs, open chromatin and RNA. Another example of guide capture in multiomic assays is ECCITE-seq[146], which combines 5′ scRNA-seq, protein detection, and targeted guide RNA capture using both reverse-transcription priming and targeted amplification.

Together, these examples highlight the versatility of targeted scRNA-seq methods. By enriching for TOIs and ROIs, they not only overcome blind spots of standard scRNA-seq but also allow integration with information on the standard transcriptome. This combination enables researchers to link specific mutations, isoforms, or microbial signals to broader transcriptional changes, expanding the scope of biological questions addressable at single-cell resolution.

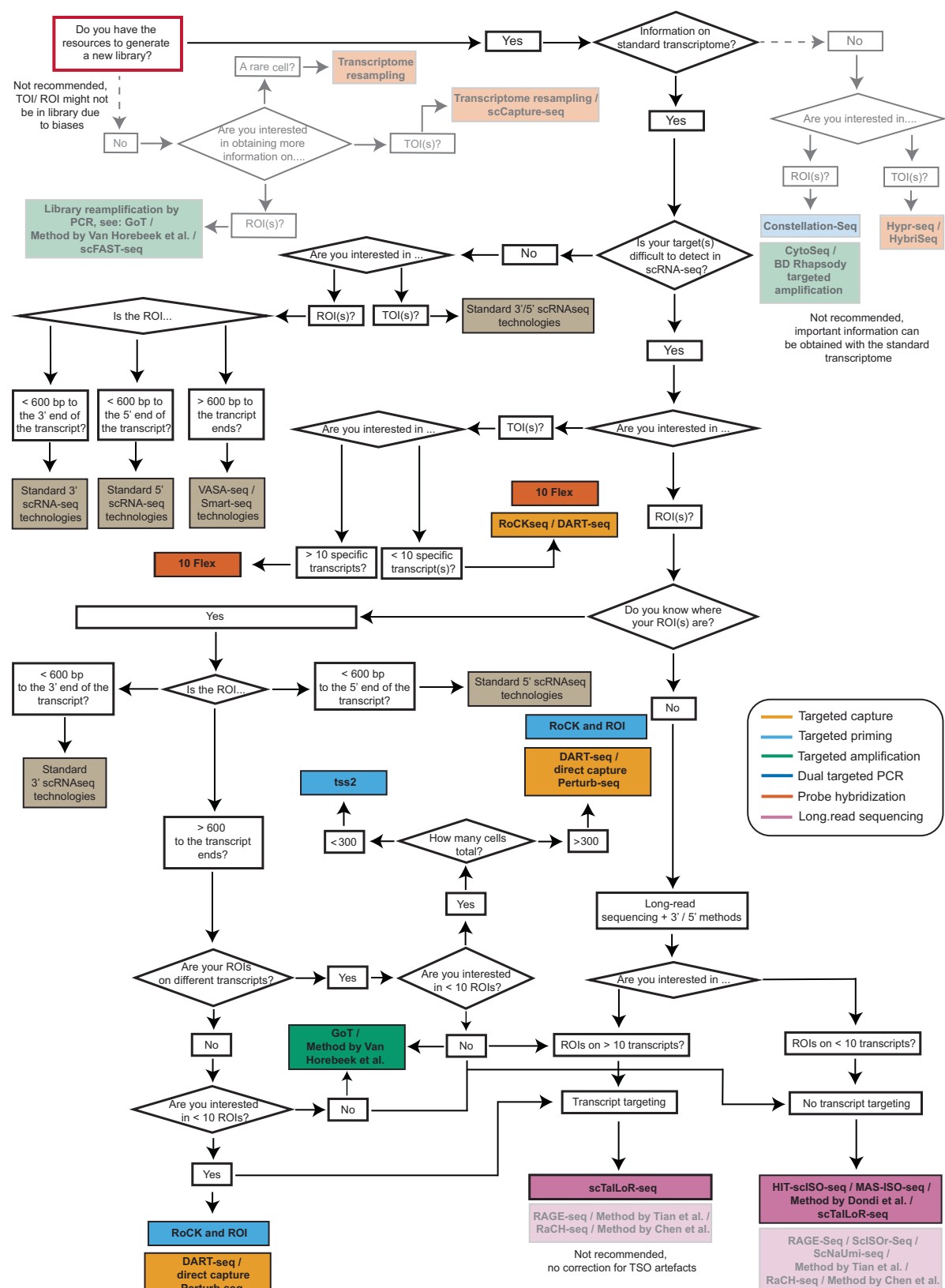

## Decision tree and points to consider when choosing a targeted scRNA-seq method

The choice of targeted sequencing method depends on the experimental question, available biomaterial, and the ability to retrieve information required to draw firm conclusions from the scRNA-seq experiment. However, the wealth of available targeting methods, the differences in targeting strategy they use, the steps in the scRNA-seq workflow, as well as the biases they tackle, makes it difficult to select the optimal strategy for a given research question. To aid with this important selection, we generated a comprehensive decision tree summarizing key considerations (Fig. 5).

**Fig. 5 | Decision tree for the choice of targeted scRNA-seq method.** Red box indicates start point for decision. Boxes and arrows in grey indicate methods which can be used for targeted sequencing but which we do not recommend due to the outlined reasons. The cutoff of 10 TOIs / ROIs was chosen somewhat arbitrarily, as the maximum number of transcripts which can be targeted depends also on the properties of the transcripts themselves. Standard scRNA-seq methods are also included, as they may suffice for the detection of TOIs and ROIs. Methods are colored based on the categories of scRNA-seq methods outlined above. Methods shown in the figure include CytoSeq[15], tss2[99], Transcriptome resampling[106], DART-seq[94], TARGET-seq[104], GoT[98], BART-Seq[103], BD Rhapsody targeted amplification[100], Direct capture Perturb-seq[96], ScNaUmi-seq[120], HyPR-seq[110], Constellation-Seq[97], scCapture-seq[107], scFAST-seq[33], Method by Van Horebeek et al.[102], scG2P[105], RoCKseq / RoCK and ROI[95], 10x Flex[109], HybriSeq[108], ScISOr-Seq[119], RAGE-Seq[118], Method by Chen et al.[126], RaCH-seq[122], MAS-ISO-seq[124], HIT-scISOseq[123], GoT-Splice[101], scTaILoR-seq[125], Method by Tian et al.[121].

While many targeted scRNA-seq methods using short-read sequencing have been described, we only included a selection based on the considerations described above. Methods requiring custom setups were excluded, as they tend to be more labor-intensive and costly. Instead, we focused on approaches that can be implemented using widely available commercial platforms.

The first node in the decision tree is whether to generate a novel library or apply targeted solutions to previously synthesized cDNAs. Wherever possible, we recommend generating a new library, as TOIs may not be captured in the existing ones. The second major node in the decision tree is the concomitant detection of the standard transcriptome, which is essential if the research question relies on the identification of cell types. Key considerations then include the number of TOIs/ROIs, the technologies available in-house, as well as the actual sequences of the targets. Because the choice of targeted method also depends on the technologies available in-house, nodes may have multiple recommended methods. An example is targeted capture methods, which are based on different commercially available technologies.

Depending on the number of targets, the 10× Flex protocol, RoCKseq and DART-seq remain the methods of choice for TOI detection as they do not introduce additional amplification and target the first step in the scRNA-seq protocol. Despite inherent PCR biases, targeted amplification remains the only solution to recover information on a high number of ROIs (>10), as multiplexing of targeted capture methods has not been extensively shown or validated.

Targeting methods described for a given platform may also be applied to a similar technology, and the methods themselves can be combined (e.g., TOI targeting for long-read sequencing technologies can be extended to most (if not all) protocols, same as technologies acting on final cDNA libraries, while targeted capture and priming can be used in the same experiment). Additionally, the methods described in Box 1, in particular those increasing information on cells of interest during sample preparation and depleting highly abundant RNAs, can also be applied in combination with targeted technologies to further enhance the detection of TOIs and ROIs.

Importantly, while we propose a set of methods based on the reasoning outlined in this review, a systematic, side-by-side comparison of all targeted scRNA-seq methods would be highly valuable for the field. Ideally, this study should be conducted on the same biological sample, preferably on the same day, to minimize batch effects. All methods should target the same TOI(s)/ROI(s), and protocols should be carried out by experts to reduce technical biases. However, such a study would be challenging, as different targeted methods are often optimized for distinct subsets of TOI(s)/ROI(s) and comparing them across divergent targets or scenarios would compromise the benchmarking objective. Our decision tree thus serves as a valuable tool to guide researchers toward both the most appropriate targeting strategy and the specific method best aligned with their research question.

## Perspectives

Currently, TOI and ROI detection in scRNA-seq experiments is limited by the fact that only a certain portion of the transcriptome is detected with current technologies. While improvements in read depth per cell (facilitated by decreased cost of sequencing) aid with the detection of TOIs, the resulting data from many such experiments is still biased towards the 3′/5′ end of transcripts. All scRNA-seq approaches still suffer from biases introduced during protocol steps such as reverse transcription and PCR. These technologies have been used for over 50 years and have already undergone multiple rounds of optimization, and it is thus unlikely that they will change. A promising solution to potentially overcome these biases is DRS, which allows bypassing the need for reverse transcription and PCR amplification. This topic has been addressed by a previous call for direct sequencing of full-length RNAs with the aim to determine modifications on RNA bases[147], but to our knowledge current methods have not been applied yet to single cells and are still associated with low throughput, high cost and challenging set ups. While further innovation in this area is expected, progress is likely to be gradual, as these approaches are currently largely conceptual.

While multiple targeting strategies have been developed, they have not been systematically assessed or compared experimentally. This benchmarking study, outlined above, would be highly relevant for the field, further crystallizing the optimal methods for TOI and ROI enhancement. The observation that targeted capture of RNAs can lead to an increase in the detection of TOIs is highly promising for future development of targeted scRNA-seq methods. Capture/priming of transcripts is the first part of scRNA-seq protocols, and biases occurring at this step are exacerbated during the downstream protocol. While this step has received relatively little attention, future improvements in scRNA-seq performance may benefit from optimizing targeted capture and priming conditions. Given the relaxed binding conditions of RNAs on barcoded beads, these solutions should focus on enhancing the specificity of capture, thus also reducing internal priming artifacts. Potential solutions could be using LNA oligonucleotides, or addition of chemicals such as formamide to reduce the melting temperature. In parallel, probe-based approaches targeting the same step, such as 10× Flex, have shown great promise. Although these methods are already highly efficient, further refinement of probe design could enhance target specificity and performance.

Another promising direction is long-read sequencing, which has seen rapid growth in the past years through the development of methods combining 3′/5′-based technologies with long-read sequencing to profile the cellular transcriptome. With the decrease in cost of long-read sequencing platforms, increased read throughput and lower error rate, the need for 3′/5′-based methods may be fully circumvented. These improvements would allow full transcripts, including ROIs, to be sequenced and the standard cellular transcriptome, including TOIs, to be profiled at the same time in single cells. Although we believe major technological advances will derive from the field of long-read sequencing, currently targeted scRNA-seq methods remain indispensable for detection of TOIs and ROIs.

Taken together, systematic improvements in targeted methods at the RNA level including targeted capture and probe hybridization, adaptation of DRS, and integration with long-read platforms represent the most promising paths to comprehensively profile TOIs and ROIs at single-cell resolution.

## Reporting summary

Further information on research design is available in the Nature Portfolio Reporting Summary linked to this article.

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

## Acknowledgements

We thank members of the Basler laboratory, including G. Hausmann, T. Dalessi, F. Fazilaty, J. Little, and T. Valenta, for their valuable input on the manuscript. Given the broad scope of this review and the wide range of papers offering diverse targeting strategies, we may not have cited all relevant work and sincerely apologize to any authors whose contributions we may have overlooked.

## Author contributions

G.M. researched the data and wrote the first draft of the manuscript. E.B. and K.B. contributed to the conception of the study and provided supervision. G.M., E.B., and K.B. contributed to discussions of the content and reviewed and edited the manuscript prior to submission.

## Competing interests

The authors declare no competing interests.
