## [Transparent Peer Review file · Communications Biology]

A practical guide to targeted single-cell RNA sequencing technologies

Corresponding Author: Dr Giulia Moro

This manuscript has been previously reviewed at another journal. This document only contains information relating to versions considered at Communications Biology.

Version 0:

Reviewer comments:

Reviewer #1

(Remarks to the Author)

I'm satisfied with the answers and how the manuscript has been revised.

Reviewer #2

(Remarks to the Author)

The authors have responded to my first-round reviewer comments and suitably addressed many of them. The manuscript has clearly improved. I have no further comments or concerns.

Reviewer #4

(Remarks to the Author)

I co-reviewed this manuscript with one of the reviewers who provided the listed reports. This is part of the Communications Biology initiative to facilitate training in peer review and to provide appropriate recognition for Early Career Researchers who co-review manuscripts.

We sincerely thank the reviewers for their thoughtful comments and constructive feedback on our manuscript. We have carefully considered each point and provide our responses below. Given the editor's guidelines, we have focused on the following points:

(1) Given the concerns from Reviewers #2 and 3 about the summary of potentially outdated techniques, it would be important to acknowledge the historical relevance of these methods, but also that they might not be practical options nowadays. This may also involve streamlining or removing the decision tree in Fig 5 to de-emphasize some of these approaches.

We agree that acknowledging which methods are relevant as of today is highly relevant, and helps guide the reader to the optimal method for their application. To address this point, we have indicated, for each targeting type, whether the strategy remains relevant today or is no longer the preferred option.

We have also streamlined the decision tree in Fig. 5 to include only the methods we currently recommend, focusing on high-throughput approaches that do not require custom setups, which are often labor-intensive and expensive. Some decision nodes still include multiple methods; we have clarified in the text that this reflects the fact that the same solution can be implemented by multiple technologies.

(2) Per point #7 from Reviewer #1, include some discussion of the relevance of these approaches to targeted spatial transcriptomics, and comparison to newer (ex. Reviewer #2, point 3, regarding 10X Genomics Flex) or other analogous methods (Reviewer #1, point #2). At the same time, it would be important to acknowledge how many cells are handled by each method (per Reviewer #3's comments).

We have included a section on spatial transcriptomics, emphasizing the importance of targeted technologies in this field.

We have included newer technologies (as the 10x Genomics Flex or HybriSeq) to the list of targeted methods, highlighting the important breakthroughs they bring to the field.

We have included a table with important features of the methods such as number of cells and recovery of the full transcriptome as a table (Table 2). This Table was previously included in Supplementary Information, but given its relevance, we have moved it to the main text.

(3) Please outline potential solutions to inherent biases in these techniques (Reviewer #1, point 6), ideally as a new display item (Box or Table). Likewise, it could be useful to include another display item to address Reviewer #2's comment (point 1) about biologically relevant applications or practical use of these methods.

We have added a box (Box 1) listing potential solutions to the described biases. These solutions are not based on targeted methods, but rather focus on general approaches to overcome the biases.

Additionally, we have added a dedicated review section on biologically relevant applications of targeted methods. As the review already contains multiple figures and tables, we chose not

to include an additional display item for this section; however, it is organized into clearly defined subchapters for easier navigation.

(4) Per comments from Reviewer #1 (point 3) and Reviewer #2 (points 2-4), please carefully review your references to ensure a balanced and (where possible) quantitative summary of the literature.

We have reviewed our references and, where available, have added quantitative information on the methods. We have not included a direct comparison of the methods in terms of enrichment efficiency and off-targets, as each method is based on a different set of TOIs / ROIs, different sequencing depth, different biological sample and different technology. These parameters could be compared thanks to an in depth benchmarking study, which is currently not available. We have detailed these considerations in the reply to Review 1, point 3.

Referee #1

The manuscript entitled “Navigating targeted single-cell RNA sequencing: a practical guide to method selection” by Moro et al. reviews methods that enable targeted single-cell RNA sequencing. After presenting an overview of the limitations and biases of non-targeted methods, the authors categorize and discuss published methods that overcome one or more of these issues. The manuscript is well written, nicely illustrated, and presents a unique and welcome addition to the field.

major comments:

1. While the authors make a laudable effort to categorize the different targeted methods, I'm not entirely convinced that the current classification is the most logical one. A) Is category ‘targeted capture’ physically ‘capturing’ RNA in the sense of physically separating or selecting transcripts? Is this not more targeting via sequence-specific reverse transcription? B) The category ‘targeted priming’ lumps together sequencing-specific reverse transcription (at RNA level), and priming at the 1st strand cDNA or (double-stranded?) cDNA level? C) ‘Targeted amplification’ and ‘targeted PCR’ are, for the average reader, the same, and only different upon reading the text. Both are PCR-based enrichment, but with one or two target-specific primers, respectively. Given the above, could the authors consider categorizing based on the workflow order (biochemical analyte) at which targeting occurs, i.e. RNA (reverse transcription), 1st strand cDNA, PCR amplified cDNA, ...?

We thank the reviewer for this insightful comment. We agree that there are multiple valid ways to categorize targeted methods and acknowledge that classification based on the biochemical analyte (e.g., RNA, first-strand cDNA, PCR-amplified cDNA) is an interesting and logical alternative. In our current framework, we chose to categorize the methods primarily by targeting strategy, with the targeted analyte presented as a secondary level (in each subsection of targeted methods). This approach aligns with the structure of the review, where the first section discusses technical biases and the subsequent sections address how different targeting strategies mitigate those biases. Categorizing primarily by analyte could indeed be another valid approach, but we opted for the current structure to maintain consistency with how technical aspects are discussed earlier in the review.

To point A: The targeted capture category indeed includes both the physical capture of transcripts of interest (TOIs) and modifications to the reverse transcription initiation step. Transcripts are first captured on barcoded beads, not only through poly(A) binding but also through sequence-specific capture oligos that hybridize to internal regions of the TOI. Reverse transcription then proceeds from these capture sites, which serve as initiation points, thus linking the targeted capture with reverse transcription starting at internal transcript regions.

2. The authors focus on positive selection of targeted transcripts or regions, but alternative strategies exist to deplete overly abundant transcripts from (single-cell) RNA-seq libraries, e.g. CRISPR based cleavage of cDNA or limited-cycle PCR amplified cDNA (e.g. ZapR in Takara kits, but also in lab-developed methods), or LNA-enhanced blocking oligonucleotides (e.g.

Everaert et al., Biol Proced Online, 2023). These methods target unwanted RNA and, hence, at least address the read depth bias.

We thank the reviewer for this comment and agree that these solutions are highly relevant. We have added a box (Box 1) to include these strategies. We focused on methods to overcome the described biases beyond targeted strategies and included depletion of highly abundant transcripts as one of the sections.

3. It would be informative if the authors could summarize the percentage enrichment, on- and off-target rates, number of UMI retrieved (at equal depth), max number of TOI/ROI detected, duplication rates, etc. for the various reviewed methods. They could also suggest a future benchmarking study to determine these metrics (in a uniform manner, using the same cells and targets).

We have moved the Supplementary Information table which included metrics on the various methods to the main section of the manuscript (Table 2). This table includes information on the platform on which the methods are based on, cellular throughput (categorized on low, medium or high), type of targeting, targeting of TOIs and/or ROIs, number of targets described and detection of the standard transcriptome. While we agree that including metrics on the percentage of enrichment and of on- and off-targets would be of interest, we believe that these metrics are not directly comparable, as each method uses different read depth, different TOIs / ROIs, different biological samples and sequences different number of cells. Additionally, the methods on which the targeted technologies are based on, as well as the sequencing technologies themselves, have greatly evolved in the past years, and comparing metrics across such different studies may be misleading. We have included quantitative information from the studies in the main text, wherever the targeted method was directly compared to the untargeted technology. However, we believe that the only solution for a global, overall comparison of the methods would be a benchmarking study, and we have outlined important features of such an endeavour in the review:

Importantly, while we propose a set of methods based on the reasoning outlined in this review, a systematic, side-by-side comparison of all targeted scRNA-seq methods would be highly valuable for the field. Ideally, this study should be conducted on the same biological sample, preferably on the same day, to minimize batch effects. All methods should target the same TOI(s)/ROI(s), and protocols should be carried out by the respective experts to reduce technical biases. However, such a study would be challenging, as different targeted methods are often optimized for distinct subsets of TOI(s)/ROI(s) and comparing them across divergent targets or scenarios would compromise the benchmarking objective. Our decision tree thus serves as a valuable tool to guide researchers toward both the most appropriate targeting strategy and the specific method best aligned with their research question.

4. The primary single-cell RNA-seq strategy appears to be bead-based methods, as mentioned in the introduction as the focus of this review paper (line 57). However, some other methods use plate-based protocols to perform single-cell barcoding (mainly in the targeted PCR and hybridisation parts of the 'Targeted scRNA-seq strategies using short-read sequencing')

section). It might be better to clarify this differentiation to the reader by adding an extra subdivision or mentioning it in the introduction.

We acknowledge that this distinction should be made clearer to the reader. We have added a statement in the introduction of the section on targeted technologies using short-read sequencing and included this information in the main text, whenever referring to plate-based methods.

5. It would be helpful for the reader to include additional considerations when selecting an appropriate method, such as cost-effectiveness, feasibility of implementation in the lab, and the quantitative metrics mentioned in major comment 3.

We thank the reviewer for this comment and have included additional considerations for choosing which targeted technology to implement. We have added these considerations at the end of each section on the targeted technologies. In general, we recommend using commercially available platforms for feasibility and ease of implementation and have indicated this in the section *Conclusions: What to consider when choosing a targeted scRNA-seq method*. As mentioned in the response to point 3, we believe that a direct comparison of the metrics described above is currently not feasible.

To aid with the choice of method, we have additionally streamlined the decision tree, including only the methods which are currently relevant.

6. While many biases and artifacts are mentioned, the authors fall short in referencing possible solutions. The PCR biases section (starting at line 171) may benefit from referencing strategies to balance the number of reads between low and high-abundance transcripts (e.g. Blomquist et al., PLoS One, 2013). Or, transcript polydenation (Smart-Seq-Total, VASA-seq) as a way to profile beyond polyadenylated genes.

As outlined in the response to point 2, we have included this information in Box1, as additional methods to address the biases above.

7. Although this is not within the scope of the review, it seems relevant to discuss the applicability of methods and findings to targeted spatial transcriptomics, with (near) single-cell resolution.

We agree that the applicability of targeted technologies to the spatial transcriptomics field is of interest. We have thus included a section in the review, distinguishing between image-based and sequencing-based technologies.

minor comments:

1. A brief introduction to the different steps in single-cell RNA sequencing and how they differ from bulk RNA sequencing may help non-expert readers. Although these steps are discussed

in the subsequent sections where biases are examined, a brief overview may be appropriate in the introduction.

We have included a short section in the introduction on the difference between bulk and single-cell RNA sequencing technologies, with the main difference being in the strategies to convey the information at the single cell level throughout the experiment.

2. Currently, only biases during sequencing library preparation are discussed. Addition of the effects of pre-analytical variables (cell dissociation, tissue storage conditions, cell fixation etc.) might make the overview of biases more complete. The authors should not elaborate, but at least briefly mention these.

We thank the reviewer for this comment, and we agree that these biases are highly relevant. We have added this information at the beginning of the section on the biases. We also included solutions to enhance the detection of cell types of interest (prior to the scRNA-seq experiment itself) in Box1. Finally, we have indicated the importance of the 10x Flex method in profiling fixed and permeabilized cells and the breakthrough that this technology has introduced.

3. Line 78/92: Do the % refer to mass, molarity, or number of genes?

These percentages refer to mass, and we have included this information to improve clarity.

4. SMARTer (now rebranded to 'SMART-Seq Total RNA-Seq Single Cell' for single cell analyses) from Verboom et al., Nucleic Acids Research, 2019 is missing from Figure 2 and line 82.

We thank the reviewer for pointing us to this method. We have included the technology in Figure 2, in the main text, as well as in Box1 for profiling of non-polyadenylated transcripts.

5. The authors may wish to define 'molecular crowding' for the non-expert reader.

We agree that this term may be misleading for the non-expert reader. We have added an additional explanation to the term molecular crowding:

Additionally, although there is a large excess of oligonucleotides on barcoded beads compared to mRNAs expressed in eukaryotic cells (10^6 - 10^7 DNA oligonucleotides vs 3 - 10×10^5 mRNAs³⁸), binding may be impaired due to steric hindrance and molecular crowding (where high numbers of macromolecules occupy most of the free space; Figure 1).

6. Line 118: Ligation of adjacent DNA oligonucleotides hybridizing to RNA is an alternative method to avoid RT bias.

We agree that ligation of adjacent DNA oligonucleotides is an important method to avoid RT bias and have included this method.

7. Line 129: I propose to add 'e.g., false fusion transcripts or circular RNAs' as examples to chimeric cDNAs

We agree with this statement and have added the proposed text to the review.

8. Line: Please specific low TS efficiency in terms of %.

While we agree that this information would be of great interest to add to the review, we have not found a convincing study indicating the percentage of TS efficiency. We have on the other hand cited the Seq-Well S³ study, where the authors implemented a randomly primed second-strand synthesis method to recover cDNA molecules where scRNA-seq TS had failed. The authors were able to increase gene detection by 10 fold compared to samples without recovery through random priming. We believe that these results highlight the importance of the template switching step (while however not directly providing a percentage of efficiency).

9. Line 157: The authors may wish to make it clearer that as a result, 5' or 3' ends of cDNA molecules (upon random priming) are not informative and should be clipped to avoid false-positive mutation calling or biased mapping.

We thank the reviewer for this comment and have added this information to the manuscript.

10. Line 224: How exactly does 'selectively extract' work? See major comment 1.

We thank the reviewer for this comment, and we acknowledge that the term is misleading. We have changed the text in the manuscript to hopefully increase clarity:

Targeted capture is based on hybridization of cellular RNAs to barcoded beads modified with oligonucleotides complementary to internal regions of targets of interest (TOIs). These capture sequences physically bind TOIs from the diverse RNA pool, enriching them prior to reverse transcription and enhancing their detection while maintaining coverage of the standard cellular transcriptome.

11. Line 245: 'second-strand': Do the authors refer to the entire 2nd cDNA strand, or the double-stranded part of the oligo?

We agree that the term second strand used in this context is misleading, as it refers to the double-stranded part of the oligo. We have fully removed this sentence from the revised version of the manuscript, as it was very detailed.

12. Line 196: The error rate % for PacBio and ONT seem outdated. Furthermore, these depend on the instrument and the flow cell. Why limit to Illumina for short-read sequencing, while alternatives exist (with lower error rates)?

We thank the reviewer for this comment and have amended the section, updating the error rates for PacBio and ONT. We have also removed references to Illumina sequencing throughout the

text (replacing them with short-read sequencing) and acknowledged that Illumina is the most common short-read sequencing platform:

The pattern of sequencing errors varies between short-read and long-read sequencing platforms such as PacBio and ONT. Although long-read technologies previously had higher error rates than short-read sequencing, recent advances have greatly improved their accuracy, with current error rates around 99% for ONT, 99.9% for PacBio, and 99.9% for Illumina (the most popular short-read sequencing platform to date).

13. Line 259: The authors may wish to explain that the off-targets originate from the relatively low temperature at reverse transcription (or hybridization), well below the T_m of the capture sequence. This is one of the reasons that PCR-based enrichment (where annealing temperature is close to the melting temperature of the primer) is more specific (less off-targeting).

We thank the reviewer for this comment, and we fully agree that the low temperature at hybridization / reverse transcription increases off-targets, while primer binding in PCR-based enrichment is more specific. We have included these considerations in the manuscript under the sections of targeted capture and targeted amplification.

14. Line 278: The text states that targeted RT priming is not possible, while Figure 2A (middle figure of targeted priming) shows that it is possible?

We agree that this information is misleading. Targeted RT priming is only possible for methods using soluble priming for RT (such as the 5' 10x Chromium technology), but not with barcoded beads, as in the latter case the dT stretch acts both as sequence for isolating polyadenylated RNAs and as RT primer. We have emphasized this in the text to hopefully convey additional clarity.

15. Line 315-316: Reverse-transcribed mRNA is cDNA, so what do the authors mean by 'cDNAs'? Double-stranded cDNA or PCR-amplified cDNA?

We agree that this sentence is misleading and have changed the text to indicate that in TARGET-seq, the first round of targeted amplification occurs on mRNAs while they are being reverse transcribed (the reagents for both reactions are added at the same time in the reaction).

16. The authors refer to oligonucleotide ligation at various instances in the manuscript, but may want to point out the known limitation of false-positive background (non-templated ligation).

We thank the reviewer for this comment and have added this information to the manuscript.

17. Line 351: First, the authors describe biotin-modified probes and primers, and then mention a 2nd type of probes, i.e. linear DNA probes. However, biotin-modified probes are also linear probes, no?

We thank the reviewer for this comment and agree that the naming of the three types of probes is misleading, as biotin-modified probes are indeed linear probes. We have changed the naming of the three probe-based targeting methods:

There are three main types of probe hybridization methods based on the type of enrichment: biotin-based enrichment, unmodified DNA probes or hairpin probe hybridization.

18. Line 372: Probe based hybridization is acclaimed not to influence 5'/3' bias, but the ligation approach selects a region of interest, no?

We agree that information is directed to internal transcript regions through probe-based hybridization methods using ligation approaches. However, as it is the probes which are sequenced and not the RNAs themselves, ROI regions cannot be properly assessed. As the statement on probe hybridization methods not influencing the 5'/3' bias is misleading, we have removed it from the text.

19. The authors may wish to indicate that different (longer) barcodes are used for long-read sequencing, compared to high-accuracy short-read read sequencing.

While we acknowledge that the barcodes used for long-read sequencing differ from those used in short-read sequencing, we have chosen not to include this detail in the review, as it falls outside the scope of the technologies being discussed.

20. Line 487: 'at' needed in this sentence?

We thank the reviewer for noticing the incomplete sentence and have amended it.

21. Figure 2B: What is the rationale for having the names at different vertical distances? For color-blind/disabled persons, thin colored lines are difficult to distinguish. Would it help to add one or more colored coded horizontal bar(s) to indicate category (for top methods) and other features (e.g. 3' end polA+, total RNA, ... for bottom methods)?

We thank the reviewer for this comment. We have included colour coded horizontal bars above each of the top methods in the figure. We have indicated if the lower methods are plate-based or bead-based.

Referee #2

This review article by Moro et al. focuses on targeted single-cell RNA sequencing, describing a range of protocol adaptations to standard single-cell RNA sequencing protocols that enable the preferential sequencing of certain RNA molecules. More specifically, the review focuses on methods that boost the recovery of a handful of RNA molecules of particularly important for a given experiment. The review does not discuss

the (arguably much larger) field of integrating multimodal readouts (e.g., proteins, perturbations) into single-cell RNA-seq assays, which is a justifiable choice that allows the current review to keep a narrow scientific focus.

Overall assessment:

- + the review provides a comprehensive documentation of prior work on the topic
- + it introduces a plausible structure that is supported by very detailed graphics
- + to my knowledge, this is the first dedicated review on the topic when defining it relatively narrowly
- the topic is relevant only to a small community, and very few of the discussed methods are widely used
- many of the described methods are of historical interest only and unlikely to be relevant for future projects
- where the review provides judgment, it often feels slightly off and/or biased by the author's own method

Suggestions for improvement:

1. The authors should add a dedicated section on biologically relevant applications and practical uses of the presented methods, either as part of the Introduction or as a final main section prior to the Discussion. It seems that the authors have applications in mind such as highly sensitive splice site detection, somatic mutation detection, and fusion protein detection (which tend to focus on one or a handful of target genes), but very few details are given. Especially when making this a dedicated section toward the end, it will be possible to provide some elaboration of how specific biological applications call for specific methods. The flowchart in Figure 15 is only useful for readers who already know quite precisely what they want, and it could be enriched by providing some example applications for each end point of the flow chart.

We thank the reviewer for this comment and have included a section on biologically relevant applications and practical uses of the methods. We focused on five different applications: non-polyadenylated transcripts, cancer mutations and fusion events, alternative splicing, rare cell types and marker genes and capture of guide RNAs.

2. Throughout the text, the authors should be more critical about the materials they refer to and provide an assessment _as of today_ which methods are practically useful and for what reasons and which applications. There are also statements recited in the text that were correct when the original papers were written, but for which the technology has advanced to a point that the statements are wrong or misleading today. For example, the statement about error rates of 13% for PacBio and 38% for ONT with a 2015 reference is far detached from today's reality. There are quite a few other points where I got the impression that the authors got it just slightly wrong, usually by taking outdated claims as universal truths. These are harder to pinpoint, and I am unfortunately not able to provide an exhaustive list of things that

should be revised. But it would be important to go step by step through the text and make sure that there is good judgment in the writing.

We thank the reviewer for this highly relevant comment. We have updated the reference on the error rates of the long-read sequencing methods. We have additionally checked the rest of the text to confirm that the stated numbers are recent and correct as of today.

3. The review seems very much written from the perspective of the authors' preprint on "RoCK and ROI" (<https://biorxiv.org/content/10.1101/2024.05.18.594120v2>) and their experience with the BD Rhapsody platform, which is a bit niche. In contrast, the authors seem to have no hands-on experience with the method that is much more widely used (10x Genomics Flex) and which arguably constitutes a new paradigm (essentially full-transcriptome capture prior to droplet encapsulation, with the option to easily add custom reagents for specific target capture) that is likely to replace many of the methods described in the manuscript for the majority of applications in human and mouse. This approach borrows from the field of spatial transcriptomics and has been shown to provide very high data quality when applied to single cells (despite the need for fixation). Speaking of fixation, I would expect that the various versions of single-cell combinatorial and combinatorial fluidic indexing should be compatible with this type of first-step hybridization approach as implemented in the 10x Genomics Flex protocol, leading to further cost reductions.

We thank the reviewer for the comment and for the thoughtful considerations on the 10x Genomics Flex protocol. We agree that this method is an extremely relevant addition of the field, and we have emphasized it in the text, as well as in the decision tree.

4. The review is largely qualitative in its assessments but fails to provide quantitative backing for the claims made. Most studies that are cited do provide benchmarking data, and while those should not be recited in an uncritical manner (they will almost always be somewhat biased toward the method used), ignoring all quantitative information is not ideal either. To the degree possible, the authors should extract quantitative performance information for each of the key methods cited (including routinely detected genes and UMIs per cell for the single-cell transcriptome and some measure of sensitivity) and provide them in tabular format, while making sure that the performance metrics they provide are indeed comparable.

We thank the reviewer for this comment and agree that quantitative benchmarking data is important when selecting a targeted method. As outlined in the response to Reviewer 1, major point 3, such study does not currently exist. We have however included quantitative data throughout the text and whenever the study provides comparison of the targeted compared to the untargeted technology. We have additionally proposed a benchmarking study to compare the quantitative metrics of the technologies.

Minor points:

Line 197: “careful optimization of these parameters” – which parameters? How to optimize them? Please elaborate.

We thank the reviewer for this comment and have amended the text as follows:

Careful optimization of sequencing depth and read length is thus essential to maximize recovery of low-abundance transcripts and reliably identify ROIs, particularly in complex single-cell datasets.

Line 400: “long-read sequencing technologies are associated with lower sequencing depth than short-read” – is this still a valid concern? It seems that combining standard 10x Genomics profiling with either Illumina or ONP sequencing, the major cost factor are the 10x Genomics reagents, not the sequencing reagents.

We thank the reviewer for this comment. We agree that in 10x Genomics profiling experiments combined with long- or short-read sequencing experiments, the major cost is often driven by the 10x reagents. Nevertheless, the statement about lower read depth with long-read technologies remains valid, as long-read information is spread across full-length transcripts. Consistent with this, Zajac et al. (2025; reference 120) compared short- and long-read single-cell sequencing and found that although short-read experiments yielded 4.5–6.5 times more reads, this did not translate proportionally to more UMIs. We have updated the manuscript to cite this study and clarify our statement:

Additionally, similar to short-read platforms and as long-read sequencing experiments typically yield lower read depth than short-read sequencing¹²⁰, TOIs need to be enriched for in the library prior to sequencing.

Line 445: “targeting early steps [...] minimizing biases” – this and other similar statements really need to be backed up by data supporting the claim. While it is plausible that early capture is superior to later enrichment (e.g., I would assume that the 10x Genomics Flex strategy is superior to the Parse Biosciences Gene Select strategy), quantitative empirical data should be cited here in support this claim (or it should be phrased much more weakly as a hypothesis/assumption rather than a claim).

We thank the reviewer for this comment and agree that these statements should ideally be supported by experimental data. As noted above, no direct comparison study of different targeting technologies, or of the specific steps targeted, is currently available. We have therefore softened the statements in the manuscript and clarified that the recommendations are based on theoretical considerations and practical experience rather than quantitative comparisons.

Line 476: the suggestion to emphasize direct RNA sequencing is purely hypothetical given the known problems (e.g., massive input material requirements) and slow progress in the field of direct RNA sequencing. More generally, the Conclusions section does not formulate

a sufficiently clear and compelling plan of where and how the field should go from here and would profit from a substantial revision/rewriting.

We thank the reviewer for this comment and have revised the perspective section accordingly. We agree that direct RNA sequencing faces significant technical challenges, such as high input material requirements, and that its application is currently largely hypothetical. However, we believe that this approach has the potential to circumvent many biases inherent in standard scRNA-seq workflows. To clarify the future direction of the field, we have summarized our perspective as follows:

Taken together, systematic improvements in targeted methods at the RNA level including targeted capture and probe hybridization, adaptation of direct RNA sequencing, and integration with long-read platforms represent the most promising paths to comprehensively profile TOIs and ROIs at single-cell resolution.

Referee #3

- I am an expert in the practicalities of single-cell sequencing, including targeting with respect to types of transcript, relative position on the gene, cell of origin and gene(s) or sequence(s) of interest. I appreciate the overview of methods and their intricacies, because it seems quite comprehensive. However, I would probably not benefit from a tool to help make decisions about which to use, for several reasons. Ultimately, such choices are constrained and not freely made across every method that has been reported.

- The methods vary between those that are directly supported by current, accessible and effective technologies or molecular biology and those that have already faded into the history of the field, and it's not yet clear to me whether the authors are able to make that distinction – ultimately much of the work may be useful as a catalogue but not for advice for the present day.

- The TOI/ROI (transcripts of interest / regions of interest) is a brave attempt to place a structure over all of the complexities and choices, positive and negative features of the methodologies, but it falls short in capturing enough meaning to enable simplification – we are still faced with multiple complex figures to pore over, papers to read, intricacies to understand.

- The work fails to adequately distinguish between the methods that work in the realm of hundreds of cells, from those that can be applied to (tens of) thousands of cells in a single experiment.

- Inevitably, there are some methods that are just easier to apply, more sensitive, more scalable and more relevant to the majority of practitioners – more likely to be successful – and the authors do not upweight these approaches in their narrative.

- A complete catalogue of methods is of some interest, especially if allied with value statements about which methods have been adopted or are simply easier / better to use. In my scientific practice, rather than basing my choice of approaches on a wide funnel, I tend to start with methods that work robustly (either for thousands or for hundreds of cells, as appropriate) and think of how to choose from a small set of adjustments to provide a solution for the problem presented. Similarly, a review and decision tool based on why a narrower set of available, robust approaches works well, would be most useful.

We thank the reviewer for the feedback. We understand that being an expert, he/she does not benefit from a tool to help make decisions about which technique to use. For non-expert readers, and in response to the useful inputs of this reviewer, we have streamlined the decision tree to include only methods that are both currently relevant and recommended. These are commercially available technologies, as custom platforms are often less accessible and more labour-intensive to implement. We have also added brief notes at the end of each of the five method sections to clarify the current applicability of the targeted approaches, along with additional recommendations in the concluding section to guide readers in selecting suitable methods. Furthermore, we moved the table on the number of

profiled cells from the Supplementary Information to the main text for greater visibility and explicitly highlighted which methods are low throughput. We hope that these revisions help focus the review on a smaller set of robust, widely usable approaches, while maintaining the necessary background and rationale for these choices.